# Intravitreal photoswitch therapy in advanced retinitis pigmentosa: a phase 1 open-label trial

Robert J. Casson [1,2] ✉, Eric Daniels[3], Christen D. Barras[4,5,6], Andrew Dwyer[5], Brian M. Strem[3], Charles C. Wykoff [7], Claudia Gregorio-King[3], Cameron Schuh[8], Richard H. Kramer [9] & Russell N. Van Gelder [10,11]

A small azobenzene photoswitch molecule (KIO-301), designed to confer light responsiveness to retinal ganglion cells, was evaluated for safety and feasibility in a first-in-human, phase 1, gene-agnostic, open-label, dose-escalation clinical trial in individuals with advanced retinitis pigmentosa (RP). KIO-301 was administered by intravitreal injection to 12 eyes of 6 participants. The primary outcome was ocular and systemic safety over 30 days. Secondary and exploratory assessments included functional vision testing, visual acuity, kinetic visual field, functional magnetic resonance imaging and participant-reported outcomes. The primary safety outcome was met, with no serious adverse events or dose-limiting toxicities observed at any point. No drug-related intraocular inflammation occurred, and all ocular adverse events were mild and procedure-related. Exploratory assessments identified variation in light perception and functional vision measures in some participants. Light-evoked blood-oxygen-level-dependent signal changes in visual cortical regions were observed following dosing and showed a temporal pattern compatible with pharmacodynamic activity. Participant-reported quality-of-life scores varied over time. In this small, nonrandomized phase 1 study in individuals with late-stage RP, intravitreal KIO-301 demonstrated an acceptable safety and tolerability profile, supporting the feasibility of photoswitch therapy in advanced RP, and motivating further evaluation in larger trials. ClinicalTrials.gov identifier: NCT05282953

RP comprises a genetically heterogeneous group of inherited retinal diseases (IRDs) characterized by gradual rod–cone photoreceptor degeneration, leading to night blindness, visual field loss and progression to profound visual impairment in most affected individuals by middle age. RP is a leading cause of blindness in the working-age population in high-income countries, with an estimated prevalence of approximately 1 in 3,000–4,000 (refs. 1–3). Hundreds of causative genes have been identified across syndromic and nonsyndromic forms[4],

underscoring the clinical appeal of gene-agnostic therapeutic strategies. Accordingly, a range of approaches are being investigated, including optoelectronic prosthetics, cell-based therapies and optogenetics[5]. We have pursued an alternative gene mutation-agnostic strategy using a small-molecule photoswitch (KIO-301) designed to render surviving retinal ganglion cells (RGCs) responsive to light[6].

Phototransduction in the vertebrate retina relies on light-induced conformational changes in retinal chromophores, ultimately enabling

**Fig. 1 | Conceptual model of photoswitch-mediated RGC photosensitization.** This schematic illustrates a hypothesis-generating conceptual framework, informed by prior preclinical studies of azobenzene photoswitches, for how KIO-301 may confer light responsiveness to RGCs in the setting of advanced photoreceptor degeneration, as described in ref. 11. **a**, In retinas with advanced photoreceptor degeneration, photoswitch molecules such as KIO-301 are proposed to gain access to RGCs via large-pore purinergic P2X7 receptors, which are functionally upregulated in rodent models of retinal degeneration.

Following cellular entry, KIO-301 is hypothesized to associate noncovalently with intracellular domains of HCN channels. In darkness, KIO-301 is predominantly in the *trans* configuration, resulting in channel block and reduced inward cation current. **b**, Upon illumination, reversible photoisomerization of KIO-301 may relieve channel block, permitting inward cation current and depolarization. The precise molecular gating mechanisms, the role of specific ion channels and the relevance of these pathways in human RP remain incompletely defined and are not implied as established by this schematic.

conversion of photon absorption into neural signals transmitted to the brain. In advanced RP, rods and cones progressively degenerate, whereas elements of the inner retina, including RGCs, may persist. Histopathological analyses of eyes from individuals with genetically diverse RP have shown that approximately two-thirds of RGCs are retained in the central retina compared to healthy controls, providing a biological rationale for therapeutic strategies targeting inner retinal neurons[7].

Synthetic photoisomerizing molecules that undergo reversible conformational change in response to light, termed photoswitches, have been developed to exploit this residual retinal circuitry[8,9]. In preclinical studies, azobenzene-based photoswitches entered RGCs and conferred light sensitivity to endogenous ion channels without the need for genetic manipulation, enabling light-driven action potential generation[6,8,10]. In murine models of advanced retinal degeneration (retinal degeneration type 1 (*rd1*) mice), the photoswitch DENAQ restored electrophysiological and behavioral responses for several days following a single intraocular injection, and a longer-acting analog, BENAQ (the active pharmaceutical ingredient in KIO-301), produced responses lasting several weeks[10]. In both mice and rabbits, BENAQ demonstrated an acceptable ocular safety profile at doses exceeding those required for photosensitization[10]. A conceptual model of the proposed mechanism of action, informed by preclinical studies[11], is given in Fig. 1.

Here we report a first-in-human evaluation of this photoswitch approach. ABACUS-1 was a phase 1, single-dose, open-label, dose-escalation study of KIO-301, administered by intravitreal injection to individuals with advanced RP.

**Table 1 | Baseline demographic and clinical characteristics of enrolled participants**

| Age (years) | Sex | Cohort | Baseline VA OD | Baseline VA OS |
|---|---|---|---|---|
| 68 | M | Cohort 1 | NLP | BLP |
| 72 | F | Cohort 1 | BLP | BLP |
| 65 | M | Cohort 1 | NLP | NLP |
| 70 | F | Cohort 2 | HM | HM |
| 69 | M | Cohort 2 | CF | HM |
| 67 | F | Cohort 2 | HM | CF |

OD, right eye; OS, left eye; VA, visual acuity.

## Results

### Participant disposition

A total of 24 individuals were prescreened between 3 November 2022 and 27 March 2023. Eighteen did not meet eligibility criteria, most commonly because visual acuity exceeded the protocol-defined threshold. Six participants met all criteria, provided informed consent and entered the study. Demographic and baseline clinical characteristics are provided in Table 1. The enrolled cohort comprised three male and three female participants, summarized in Table 1. Three participants with no light perception (NLP) or bare light perception (BLP) were assigned to cohort 1, and three participants with hand-motion (HM) or count-fingers (CF) vision were assigned to cohort 2.

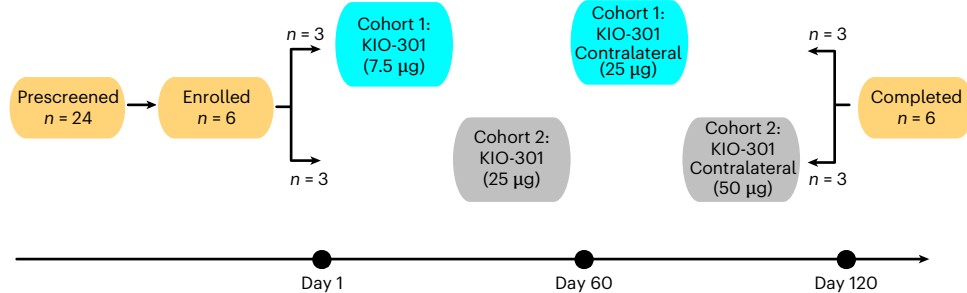

**Fig. 2 | Study design and participant disposition.** Twenty-four individuals were prescreened; six met eligibility criteria and were enrolled. Participants were assigned to cohort 1 or cohort 2 based on baseline visual function and received unilateral intravitreal administration of KIO-301 followed by protocol-specified contralateral dose escalation after safety review. The timing of dosing and follow-up assessments is shown schematically. Safety monitoring, including adverse-event surveillance, occurred continuously throughout the study period, with the prespecified primary safety endpoint assessed at day 30 following each intravitreal administration. All enrolled participants completed the study, with no early discontinuations.

The overall study design, cohort allocation, dose-escalation sequence and timing of assessments are summarized in Fig. 2. KIO-301 was administered as a single intravitreal injection to the right eye during part 1 of the study (7.5 μg for cohort 1; 25 μg for cohort 2). Following safety review, the contralateral eye was treated during part 2, with cohort 1 receiving 25 μg and cohort 2 receiving 50 μg. All six participants received the planned initial intravitreal dose of KIO-301 followed by the protocol-specified contralateral higher dose, and all participants completed the day 30 primary endpoint visit, with no early discontinuations and no missing primary endpoint data.

## Primary outcome: safety and tolerability

The primary safety outcome was met. No serious adverse events or dose-limiting toxicities were observed at any dose level. Further, no drug-related systemic adverse events were reported. Vital signs, hematology, serum biochemistry and electrocardiographic parameters remained within normal limits throughout the study.

Of the reported ocular adverse events, all were mild and transient. One participant experienced mild peri-injection discomfort and eyelid swelling following administration of 25 μg of KIO-301, deemed unrelated to study drug. A second participant experienced a mild increase in intraocular pressure following administration of 7.5 μg of KIO-301, in the context of borderline elevated baseline intraocular pressure; this event was managed with topical therapy and resolved without sequelae. No participant developed intraocular inflammation, vitreous haze, macular edema or treatment-related structural retinal changes on fundus examination, fundus autofluorescence or optical coherence tomography. Structural ocular safety assessments demonstrated findings typical of advanced RP without evidence of treatment-related structural abnormalities (Extended Data Fig. 1). A complete listing of ocular adverse events is provided in Table 2.

## Secondary outcomes

Given the first-in-human, safety-focused design and small sample size, all functional vision and neuroimaging outcomes are presented descriptively as exploratory pharmacodynamic observations. No analyses were prespecified to formally test efficacy hypotheses.

**Light perception.** Light perception was assessed using a repeated forced-choice task. Individual eye-level data are shown in Extended Data Fig. 2.

In cohort 1, which comprised participants who were NLP or BLP at baseline, variability in task performance was observed across study visits, with some participants demonstrating nonzero responses at post-treatment assessments. One participant with longstanding NLP (>10 years) reported subjective awareness of light perception within 2 days of treatment; this experience was documented in a post-study interview (Supplementary Video 1). Participants in cohort 2 had higher baseline light perception performance, with no consistent change in this assessment observed following treatment.

**Visual acuity.** Visual acuity was assessed using the Berkeley Rudimentary Vision Test (BRVT) and recorded as logarithm of the minimum angle of resolution (logMAR), with lower values indicating better

**Table 2 | Treatment-emergent ocular adverse events following intravitreal KIO-301 administration[a]**

| MedDRA term | KIO-301 | | | Severity | Drug-related |
|---|---|---|---|---|---|
| | 7.5 μg, n=3 (%) | 25 μg, n=6 (%) | 50 μg, n=3 (%) | | Total, n=12 (%) |
| Eye pain | 0 (0) | 2 (33) | 0 (0) | Mild | Unlikely 2 (17) |
| Eye swelling | 0 (0) | 1 (17) | 0 (0) | Mild | Unlikely 1 (8.3) |
| Ocular hypertension | 1 (33) | 0 (0) | 0 (0) | Mild | Possible 1 (8.3) |
| Anterior chamber cell | 0 (0) | 0 (0) | 0 (0) | NA | NA 0 (0) |
| Anterior chamber flare | 0 (0) | 0 (0) | 0 (0) | NA | NA 0 (0) |
| Vitreous cells | 0 (0) | 0 (0) | 0 (0) | NA | NA 0 (0) |
| Retinitis | 0 (0) | 0 (0) | 0 (0) | NA | NA 0 (0) |
| Vasculitis | 0 (0) | 0 (0) | 0 (0) | NA | NA 0 (0) |
| Iritis | 0 (0) | 0 (0) | 0 (0) | NA | NA 0 (0) |
| Keratic precipitates | 0 (0) | 0 (0) | 0 (0) | N/A | NA 0 (0) |
| Photophobia | 0 (0) | 0 (0) | 0 (0) | NA | NA 0 (0) |
| Photopsia | 0 (0) | 0 (0) | 0 (0) | NA | NA 0 (0) |
| Vitreous floaters | 0 (0) | 0 (0) | 0 (0) | NA | NA 0 (0) |
| Punctate keratitis | 0 (0) | 0 (0) | 0 (0) | NA | NA 0 (0) |
| Conjunctival hyperemia | 0 (0) | 0 (0) | 0 (0) | NA | NA 0 (0) |

[a]No moderate or severe ocular adverse events were observed at any dose level. MedDRA, *Medical Dictionary for Regulatory Activities*; NA, not applicable.

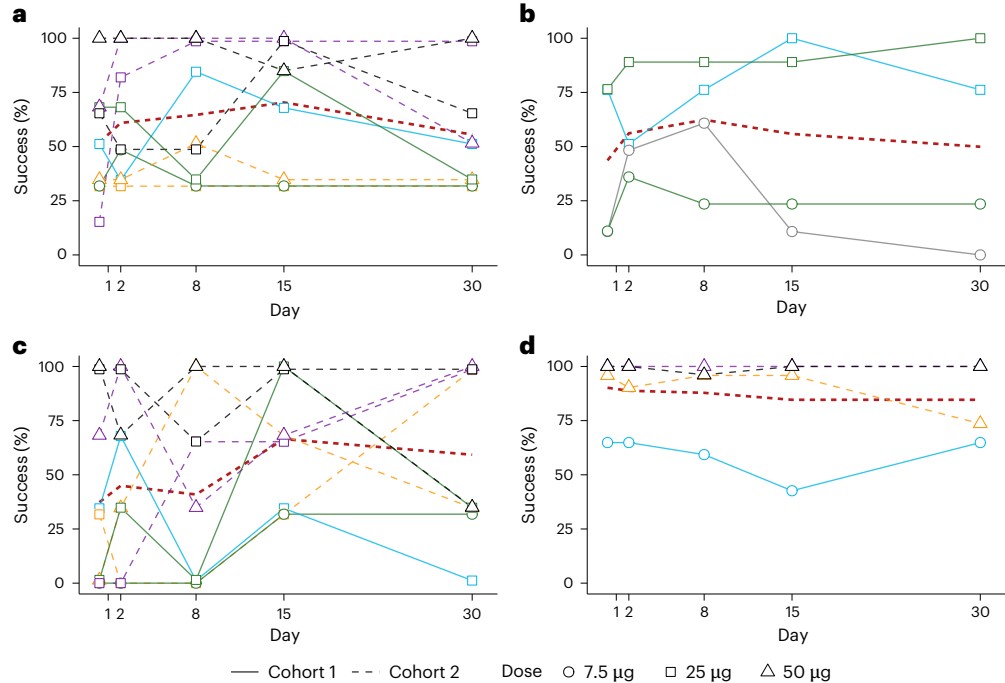

**Fig. 3 | Functional vision task performance following intravitreal KIO-301 administration. a–d,** Time courses showing individual eye-level performance on functional vision tasks. A dotted red line indicates the mean trajectory across all plotted observations and is shown solely as a visual guide to aid interpretation of individual-level data; no statistical inference was performed. Tasks included walking direction (**a**), window location (**b**), navigation (**c**) and door location (**d**). Assessments were performed under controlled ambient illumination conditions: window location at high-level illumination (350 lux), door location at mid-level illumination (150 lux) and navigation and walking direction tasks averaged across illumination levels (45–350 lux). Each line represents an individual treated eye; colors denote individual participants and symbol shapes indicate dose level (7.5, 25 or 50 µg). Small vertical offsets were applied to overlapping data points for visual clarity.

acuity (Methods and Supplementary Methods). Visual acuity assessment after screening using the BRVT was introduced as a protocol amendment and was therefore performed at baseline and day 29–30 only. At this level of profound visual impairment, BRVT-derived logMAR values reflect performance on simplified spatial vision tasks rather than recognition of conventional optotypes. In cohort 1, visual acuity could not be quantified on this scale because of the severity of retinal degeneration and was recorded as logMAR > 2.9 at all time points.

In cohort 2, individual eye-level trajectories are shown in Extended Data Fig. 3a. These values are presented descriptively in the context of known variability in visual acuity testing among individuals with ultra-low vision.

**Kinetic visual field.** Kinetic visual fields were assessed using manual Goldmann perimetry and quantified as horizontal field extent (Methods). Individual eye-level trajectories are shown in Extended Data Fig. 3b. Changes in horizontal field extent varied substantially between eyes, with heterogeneous trajectories observed over the 30-day follow-up period.

**Functional vision.** Functional vision was assessed using a battery of orientation and mobility tasks under controlled lighting conditions (Fig. 3). Across participants, performance on the walking direction task showed temporal variation over the follow-up period, with values near chance level at baseline, higher values observed at intermediate time points (peaking at day 15) and a return toward baseline by day 30 (Fig. 3a). In cohort 1, performance on the window location task varied over time, with the highest proportion of successful trials observed at day 7 (Fig. 3b). Performance on the room exit task also varied across time points, with higher values observed at day 14 followed by lower values at day 30 (Fig. 3c). No consistent temporal pattern was observed for the door location task (Fig. 3d).

**Functional magnetic resonance imaging.** Qualitative functional magnetic resonance imaging (fMRI) blood-oxygen-level-dependent (BOLD) signal maps showed suprathreshold stimulus-associated signal following KIO-301 administration (Fig. 4). Across participants, signal was observed in multiple cortical regions, including in occipital cortex encompassing primary visual cortex (V1) and adjacent extrastriate areas, at early post-treatment time points.

Visually evoked cortical signal was most prominent within 2–3 days following treatment and showed reduced spatial extent at later time points. These observations are exploratory and descriptive and were not designed to establish visual perception or treatment efficacy.

**Participant-reported outcomes.** Vision-related quality-of-life scores showed modest within-participant changes with substantial interindividual variability (Extended Data Fig. 4).

**Pharmacokinetics.** Plasma concentrations of KIO-301 were below the lower limit of quantification (0.2 ng ml⁻¹) in all participants at 4 h and 14 days following intravitreal administration, except for one participant receiving 50 µg in whom a low plasma concentration was detected at 4 h post-dose. No accumulation was observed.

## Integration of exploratory functional and neuroimaging findings

In several participants, changes in functional vision task performance and self-reported light-evoked sensory experiences were observed at early post-treatment time points following KIO-301 administration. In addition, stimulus-associated BOLD signal was detected in occipital cortical regions on fMRI at corresponding visits. These observations were heterogeneous across participants and were most prominent within the first 2–3 days following treatment, with reduced spatial extent and task performance at later time points.

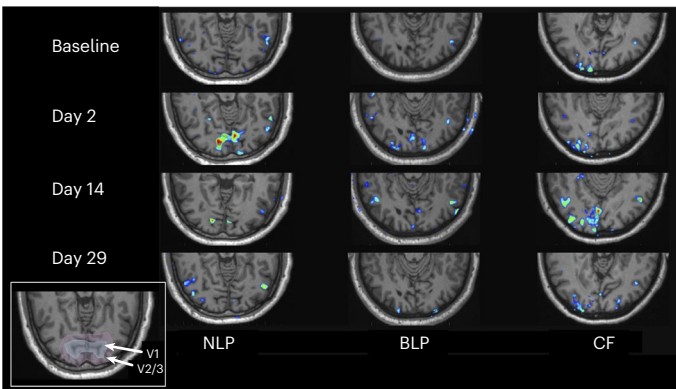

**Fig. 4 | fMRI BOLD signal maps following intravitreal KIO-301 administration.** Representative axial fMRI slices from three participants showing voxel-wise BOLD signal maps overlaid on structural images at baseline and post-treatment time points. Images are shown for one participant with NLP, one with BLP and one with CF vision at baseline. Colored voxels represent suprathreshold task-associated BOLD signal displayed relative to baseline under the same stimulus conditions. The inset illustrates the anatomical location of striate cortex (V1) and adjacent extrastriate cortex (V2/V3).

## Discussion

There is an urgent unmet need for safe and cost-effective vision restoration approaches for individuals affected by IRDs and other outer retinal degenerative diseases such as age-related macular degeneration. KIO-301 is an intravitreal formulation of the photoswitch molecule, BENAQ, which in preclinical studies was shown to selectively enter RGCs downstream of degenerated photoreceptors and render endogenous voltage-gated cation channels sensitive to light without requiring genetic manipulation; thus, enabling photocontrol of optic nerve action potential generation in response to light[6,8,10].

In this first-in-human clinical trial of any photoswitch molecule, intravitreal KIO-301 demonstrated a favorable safety and tolerability profile. There were no serious adverse events or definite drug-related adverse events. In contrast to some viral vector-based gene therapy approaches for IRDs, intraocular inflammation was not observed[12], and no structural retinal changes were detected on clinical fundus examination, fundus autofluorescence or optical coherence tomography. Reported adverse events were mild, transient and consistent with known effects of intravitreal injection procedures, including transient discomfort and mild elevations in intraocular pressure. Given the widespread clinical use of repeated intravitreal injections for retinal diseases such as neovascular age-related macular degeneration, these findings support the feasibility of intravitreal delivery of photoswitch therapy in individuals with advanced RP.

Although safety was the primary objective of the ABACUS-1 trial, the study also provided early insights into the feasibility and performance of functional, imaging and participant-reported assessments in a population with profound visual impairment. These data inform the selection and refinement of outcome measures for future studies and highlight the challenges inherent in assessing visual function at very low levels of residual vision. In addition, the presence of task-associated cortical signal in visual cortical regions following dosing may represent an exploratory downstream pharmacodynamic response compatible with target engagement.

Preclinical studies suggest that second-generation photoswitch molecules such as BENAQ selectively target RGCs in regions of outer retinal degeneration while sparing intact retinal circuitry[8]. BENAQ is proposed to enter RGCs via P2X7 channels expressed on the cell surface[11] (Fig. 1). In rodent models of photoreceptor degeneration, P2X7 receptor expression is increased and localizes to the inner retina, including the RGC layer[13,14]. In humans, single-cell RNA sequencing of the normal adult retina demonstrates low-level, heterogeneous P2X7

expression in subsets of RGCs[15], and it remains unknown whether similar degeneration-associated upregulation occurs in RP.

Hyperpolarization-activated cyclic nucleotide-gated (HCN) channels contribute to retinal signaling in mammalian models[16], and human retinal single-cell transcriptomic datasets demonstrate low-level, heterogeneous expression of HCN1 and HCN4 transcripts in RGC populations[15]. Consistent with this, ivabradine (a selective HCN channel inhibitor) is associated with reversible visual disturbances in humans, suggesting that modulation of retinal HCN channel activity can influence visual perception[17]. Preclinical studies suggest that BENAQ-mediated responses involve HCN channels, although the underlying molecular and structural gating mechanisms remain unclear[8].

In mice, photoswitches also appear to suppress the 5–10 Hz rhythmic noise in RGC firing that occurs after outer retinal degeneration[18], which would be expected to improve signal-to-noise and visual acuity in both injected and potentially contralateral eye vision. Such an effect has been noted following unilateral gene therapy for Leber hereditary optic neuropathy[19]; however, such mechanisms remain speculative in the current study.

This study has several important limitations. The sample size was necessarily small, consistent with a first-in-human, phase 1 dose-escalation design, limiting the ability to draw quantitative or generalizable conclusions. The open-label, single-arm nature of the trial and the absence of a control group preclude causal inference and do not allow efficacy to be assessed. Accordingly, all functional, imaging and participant-reported outcomes in this study are exploratory and descriptive, and were not designed or powered to support efficacy inference. The duration of follow-up was short relative to the anticipated pharmacodynamic time course of photoswitch activity and was not designed to assess durability of effect. In addition, although the light intensity used in this study was selected on the basis of preclinical data and conservative ocular safety considerations, the relationship between photon flux, retinal engagement and behavioral response in humans remains uncertain and was not optimized in this first-in-human, safety-focused study. Full-field stimulus testing, although valuable for quantifying global retinal light sensitivity, was not included as a prespecified outcome because this study was not designed to establish irradiance thresholds or efficacy endpoints. Finally, although preclinical studies provide a biologically plausible framework for photoswitch-mediated RGC photosensitization, direct mechanistic confirmation in human retinal tissue was not feasible in this study.

In this first-in-human phase 1 study, intravitreal KIO-301 was well tolerated in individuals with advanced RP, with no dose-limiting toxicities or structural retinal adverse effects observed over the prespecified safety period. Consistent with preclinical data, any photoswitch-mediated effects observed were transient, and the study was not designed to assess durability, repeat dosing or clinical efficacy.

Taken together, the temporally aligned participant-reported light-evoked visual sensations, functional task performance and stimulus-associated cortical activation observed in a subset of participants are compatible with an exploratory pharmacodynamic effect of intravitreal KIO-301 in humans. These findings extend prior preclinical and translational work on small-molecule azobenzene photoswitches as light-activated pharmacologic agents and support the feasibility of this approach for further clinical investigation[20]. The relevance of these findings lies in establishing the feasibility and ocular safety of intravitreal photoswitch delivery in humans and in demonstrating signals compatible with pharmacodynamic target engagement. These data provide a necessary translational foundation for future, appropriately powered studies to evaluate repeat administration strategies, optimization of stimulation parameters and potential clinical utility.

## Online content

Any methods, additional references, Nature Portfolio reporting summaries, source data, extended data, supplementary information,

acknowledgements, peer review information; details of author contributions and competing interests; and statements of data and code availability are available at https://doi.org/10.1038/s41591-026-04317-6.

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

¹Department of Ophthalmology, Royal Adelaide Hospital, Adelaide, South Australia, Australia. ²Discipline of Ophthalmology & Visual Sciences, Adelaide University, Adelaide, South Australia, Australia. ³Kiora Pharmaceuticals, Encinitas, CA, USA. ⁴Department of Radiology, Royal Adelaide Hospital, Adelaide, South Australia, Australia. ⁵South Australian Health and Medical Research Institute, Adelaide, South Australia, Australia. ⁶Faculty of Health and Medical Sciences, Adelaide University, Adelaide, South Australia, Australia. ⁷Retina Consultants of Texas, Retina Consultants of America, Blanton Eye Institute, Houston Methodist Hospital, Houston, TX, USA. ⁸Ora Inc., Andover, MA, USA. ⁹Department of Neuroscience, University of California, Berkeley, CA, USA. ¹⁰Karalis-Johnson Retina Center, Department of Ophthalmology, University of Washington School of Medicine, Seattle, WA, USA. ¹¹Departments of Neurobiology & Biophysics and Laboratory Medicine & Pathology, University of Washington School of Medicine, Seattle, WA, USA. ✉e-mail: robert.casson@adelaide.edu.au

## Methods

### Study design and oversight

ABACUS-1 was a first-in-human, open-label, phase 1 dose-escalation study designed to evaluate the safety and feasibility of intravitreal administration of the photoswitch molecule KIO-301 in individuals with advanced RP. The study was conducted at two clinical sites in Adelaide, Australia. The protocol was approved by the Central Adelaide Local Health Network Human Research Ethics Committee and registered on ClinicalTrials.gov (NCT05282953). The study was conducted in accordance with the Declaration of Helsinki and the International Conference on Harmonisation Good Clinical Practice guidelines. Written informed consent was obtained from all participants before enrollment. Participants received a stipend to cover travel and meal expenses. The study was conducted between November 2022 and September 2023, with follow-up completed after the final study visit for the last enrolled participant. The study protocol and statistical analysis plan are available from the corresponding author upon request and are also accessible via ClinicalTrials.gov (NCT05282953).

### Participants

Male, female and nonbinary participants were eligible for inclusion, and no sex-based exclusion criteria were applied. Gender was self-identified. Participants were adults aged 18 to 80 years with a clinical diagnosis of advanced RP and severe visual impairment, defined as NLP, BLP, HM or CF vision.

Exclusion criteria included clinically significant ocular comorbidity, active ocular inflammation or infection, prior retinal detachment or any condition that, in the investigator's judgment, could compromise safety or study participation.

Full inclusion and exclusion criteria are provided in the Supplementary Methods. Participants were enrolled sequentially and assigned to dose cohorts according to the predefined dose-escalation scheme. Recruitment and study information were facilitated with the assistance of a patient support organization (Retina Australia).

### Intervention and dose escalation

KIO-301 was administered as a single 50-μl intravitreal injection. Dose escalation was conducted in two sequential parts. In part 1, participants received KIO-301 in the right eye at either 7.5 μg (cohort 1) or 25 μg (cohort 2). Following review of safety data, part 2 involved treatment of the contralateral eye, with cohort 1 receiving 25 μg and cohort 2 receiving 50 μg. Advancement to higher dose levels was contingent on the absence of dose-limiting ocular or systemic adverse events and was based solely on predefined safety criteria, not on functional or imaging outcomes.

### Preclinical validation of photoswitch activity

Before clinical administration, the biological activity of KIO-301 was confirmed using a retinal explant model. All animal procedures were approved by the University of California, Berkeley Institutional Animal Care and Use Committee (protocol AUP-2016-04-8700-3) and conducted in accordance with the National Institutes of Health Guide for the Care and Use of Laboratory Animals. Mice were maintained on a 12:12 h light–dark cycle under controlled temperature (21 °C) and humidity.

KIO-301 was prepared as a 10 mM stock solution in DMSO and diluted to a working concentration of 100 μM in physiological saline. Retinas from *rd1* mice aged postnatal day 30–60 were isolated and incubated with KIO-301 for 30 min at 21 °C, followed by washing in drug-free saline. Treated retinas were mounted ganglion cell side down on a 60-electrode multielectrode array (Multi Channel Systems) for extracellular recording of RGC activity. Light stimulation was delivered using 1-s flashes from a 455 nm LED light source at approximately 1.5 mW cm$^{-2}$. Untreated rd1 retinas served as negative controls.

Light-evoked responses were quantified using a photoswitch index, defined as the difference between mean firing rate during light stimulation and mean firing rate in darkness, divided by the sum of firing rates during light and dark conditions. Representative recordings are shown in Extended Data Fig. 5. These experiments were performed to confirm biological activity of the administered compound before clinical dosing and were not designed to assess therapeutic efficacy.

### Influence of preclinical data on trial design

Preclinical retinal explant and in vivo studies demonstrated that azobenzene photoswitch compounds confer rapid, reversible light responsiveness to RGCs at low micromolar concentrations, with biological activity observed at doses substantially below those associated with ocular toxicity in animal models. These findings informed the selection of a conservative starting dose for first-in-human evaluation and supported a safety-led dose-escalation strategy rather than targeting a putative efficacy threshold. Preclinical studies also indicated that photoswitch-mediated responses occur rapidly following intraocular administration and decline over days to weeks, consistent with a short-acting pharmacodynamic profile. Accordingly, the clinical study design incorporated early post-dosing assessments to capture potential transient functional and cortical responses, while prioritizing safety and tolerability as the primary outcome measures.

Preclinical studies demonstrated photoswitch-mediated retinal responses across a broad range of photon fluxes under controlled experimental conditions in which retinal illumination could be estimated directly. By contrast, retinal photon flux cannot be directly measured in vivo in human participants; therefore, light stimulation parameters in this first-in-human study were selected conservatively based on incident illumination and ocular safety considerations and corresponded with preclinical levels toward the lower end of photoresponsiveness.

### Clinical assessments

**Safety.** Participants underwent comprehensive ocular and systemic safety assessments at each study visit. Ocular assessments included slit-lamp biomicroscopy, intraocular pressure measurement, dilated fundus examination, fundus color photography, retinal autofluorescence imaging and spectral-domain optical coherence tomography. Systemic assessments included vital signs, hematology, biochemistry, and electrocardiography. Adverse events were recorded at each visit and coded using the Medical Dictionary for Regulatory Activities[21].

### Visual function assessments

**Light perception.** Light perception was assessed using a repeated forced-choice paradigm in which participants were asked to identify the presence or absence of an illuminated target presented on a dark background across multiple trials. Stimuli were presented monocularly to the treated eye. Details are provided in the Supplementary Methods. The proportion of correct responses was recorded at each study visit.

**Visual acuity.** Visual acuity was assessed using the BRVT, which is validated for individuals with profound visual impairment[22]. Visual acuity was recorded in logMAR units where measurable, with predefined criteria applied for NLP, BLP, HM and CF vision. Testing was performed monocularly under standardized lighting conditions by trained examiners. Details are provided in the Supplementary Methods.

**Kinetic visual fields.** Kinetic visual fields were assessed using manual Goldmann perimetry with a blue light stimulus (440–460 nm). Testing was performed monocularly using standardized protocols. Visual field extent was quantified as the summed horizontal field extent across meridians.

**Functional vision testing.** Functional vision was evaluated using a battery of standardized orientation and mobility tasks performed under controlled lighting conditions. Tasks included walking direction, window location, room exit and door location tests. Tasks were selected to assess navigation and spatial orientation in participants

with severe visual impairment. Visual performance was assessed under predefined illumination levels (45, 125 and 350 lux). For presentation, results are shown at the illumination level(s) most informative for each task, based on task design and observed dynamic range. Performance was recorded as the proportion of successful trials for each task at each visit. Details are provided in the Supplementary Methods. In brief, the tests comprised the following assessments:

- Walking direction test: Participants were instructed to determine the direction of travel indicated by a visual cue. A trial was scored as successful if the participant correctly identified the direction of movement.
- Window location test: Participants were asked to identify the location of a window positioned in the testing environment. A trial was scored as successful if the participant correctly identified the window location.
- Room exit test: Participants were instructed to navigate through a room and locate the exit. A trial was scored as successful if the participant exited through the correct doorway without assistance.
- Door location test: Participants were asked to locate and navigate to a door positioned in the testing environment. A trial was scored as successful if the participant reached the correct door.

For each task, performance was recorded as the proportion of successful trials per visit.

### fMRI
fMRI was performed to assess cortical responses to visual stimulation. Imaging was conducted using a clinical MRI scanner with BOLD contrast. Visual stimuli, including flickering checkerboard and on/off paradigms, were delivered monocularly to the treated eye during image acquisition. Preprocessing steps included motion correction, spatial normalization and smoothing. Imaging analyses were descriptive in nature and were not designed for confirmatory hypothesis testing. Full acquisition parameters, stimulus timing, preprocessing pipelines and region-of-interest definitions are provided in the Supplementary Materials. Image review and post-processing were performed using Siemens Syngo Via (Siemens Healthineers) and Nordic NeuroLab software as detailed in the Supplementary Methods.

### Quality of life
Participant-reported outcomes were assessed using the National Eye Institute Visual Function Questionnaire-25 (ref. 23). Composite scores were calculated according to published scoring guidelines.

### Pharmacokinetics
Plasma samples were collected at predefined timepoints following intravitreal administration of KIO-301. Plasma concentrations were measured using a validated analytical assay with a lower limit of quantification of 0.2 ng ml$^{-1}$.

### Statistical considerations
This study was designed as a first-in-human, phase 1 investigation with the primary objective of evaluating safety and tolerability. Accordingly, no formal hypothesis testing was prespecified, and the sample size was not determined by statistical power calculations. Rather, cohort size was selected pragmatically to enable careful safety evaluation across a limited number of participants, consistent with early-phase ophthalmic clinical studies and ethical considerations surrounding exposure to an investigational therapy.

All analyses were descriptive. Continuous and categorical outcomes are presented at the individual participant level without formal inferential testing. Functional, imaging and participant-reported measures were evaluated to characterize variability and potential pharmacodynamic signals, and to inform the design of subsequent studies, rather than to test predefined efficacy hypotheses.

### Inclusion and ethics statement
This was an industry-sponsored trial conducted in a high-income country (Australia).

### Reporting summary
Further information on research design is available in the Nature Portfolio Reporting Summary linked to this article.

### Data availability
De-identified participant-level data supporting the findings of this study are not publicly available due to institutional ethics restrictions, sponsor confidentiality obligations and the small sample size of this first-in-human clinical study, which may increase re-identification risk. The minimum dataset necessary to interpret, verify and extend the findings reported in this article is available from the corresponding author upon reasonable request. Requests for access should be directed to the corresponding author (email: robert.casson@adelaide.edu.au) and will be considered subject to institutional review board or ethics approval where required, sponsor review and execution of an appropriate data-use or confidentiality agreement. Approved data may be used solely for noncommercial academic research consistent with the original informed consent and ethics approvals and may not be redistributed. Requests will be acknowledged within one week and a decision provided within two weeks.

### Code availability
No custom code was developed for this study. Data processing and analysis were performed using standard, commercially available and/or widely used statistical software as described in the Methods.

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

### Acknowledgements
This study was funded by Kiora Pharmaceuticals, Inc., which provided the investigational product KIO-301 and financial support for the conduct of the trial. The sponsor contributed to study design and interpretation of the results through authorship but had no role in patient recruitment or delivery of clinical care. R.N.V.G. acknowledges support from an Unrestricted Departmental Grant from Research to Prevent Blindness and the Mark J. Daily, MD Research Fund. We thank S. Sperl and L. Plasser for overseeing the manufacturing of the drug substance and drug product used in the studies described in this manuscript. We also acknowledge L. Graetz and M. Jenkinson for their contributions to the analysis of quantitative functional magnetic resonance imaging data. The authors acknowledge A. Walls and the facilities of the National Imaging Facility, a National Collaborative Research Infrastructure Strategy (NCRIS) capability, at the South Australian Health and Medical Research Institute. We thank L. Tuomi for assistance with manuscript preparation. We also acknowledge Retina Australia for assistance with participant identification and recruitment.

### Author contributions
R.J.C., E.D., B.M.S., C.D.B., A.D. and R.N.V.G. conceived and designed the study. R.J.C. was responsible for patient recruitment and clinical

procedures. E.D., R.J.C., B.M.S., C.D.B., A.D., R.H.K. and R.N.V.G. acquired, analyzed and interpreted the data. R.J.C. and E.D. drafted the manuscript. E.D., B.M.S., C.D.B., A.D., C.C.W., C.G-K., C.S., R.H.K. and R.N.V.G. critically reviewed and edited the manuscript. R.J.C. and R.N.V.G. supervised the study. All authors approved the final version of the manuscript.

## Funding

## Competing interests

E.D. and B.M.S. are employees and officers of Kiora Pharmaceuticals, Inc., and hold equity in the company. C.G-K. is an employee of Kiora Pharmaceuticals, Inc., and holds equity in the company. C.C.W. and R.H.K. serve as consultants to Kiora Pharmaceuticals, Inc. C.S. is an employee of Ora, Inc. The other authors declare no competing interests.

## Additional information

**Extended data** is available for this paper at

**Correspondence and requests for materials** should be addressed to Robert J. Casson.

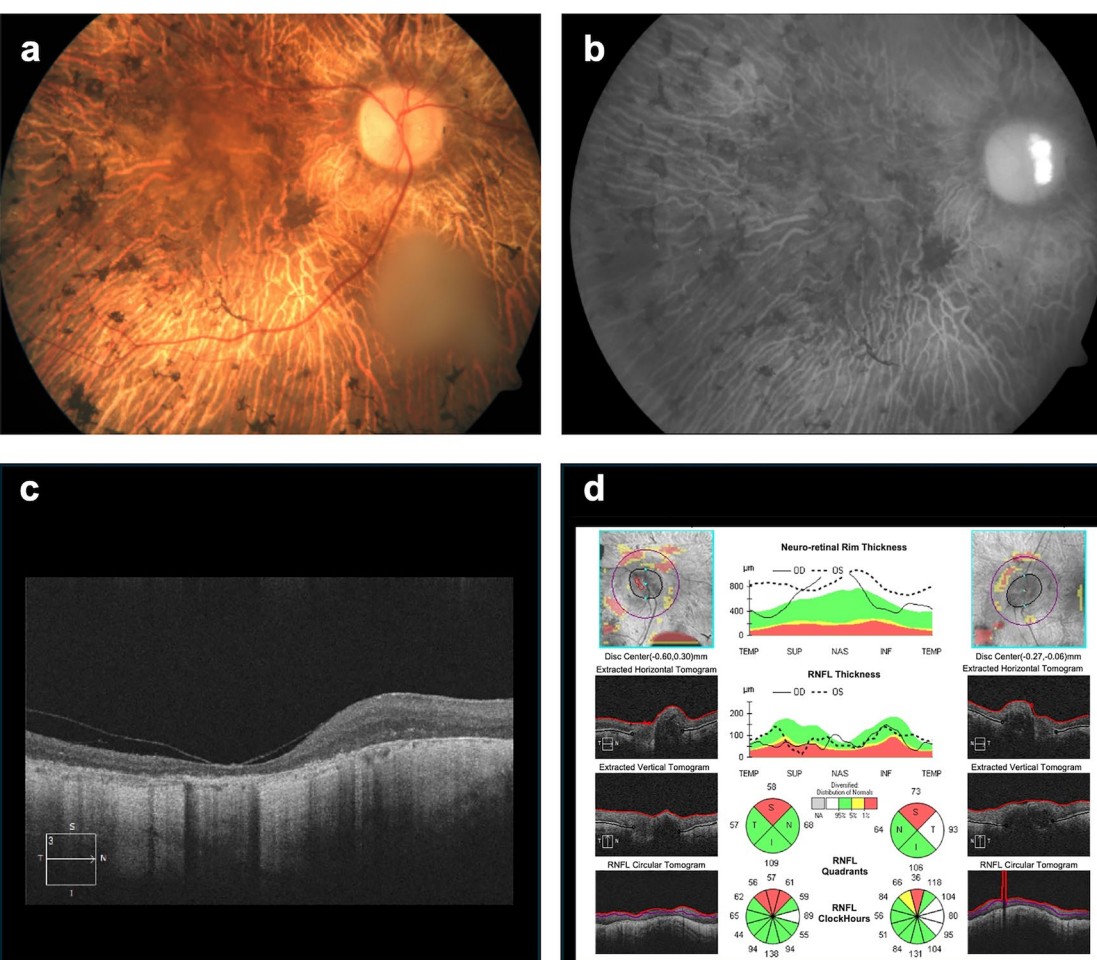

**Extended Data Fig. 1 | Retinal imaging in advanced retinitis pigmentosa. (a–d)** Representative retinal imaging from a study participant with advanced retinitis pigmentosa. These imaging assessments were performed as part of protocol-defined ocular safety monitoring and were not intended as efficacy endpoints. No treatment-related structural retinal abnormalities were observed. (**a**) Color fundus photograph demonstrating characteristic features of advanced disease, including bone-spicule pigmentation, attenuated retinal vessels, and optic disc pallor. (**b**) Fundus autofluorescence image showing widespread loss of autofluorescence consistent with severe photoreceptor degeneration. (**c**) Spectral-domain optical coherence tomography (OCT) of the macula illustrating marked outer retinal thinning with relative preservation of inner retinal layers. (**d**) OCT-derived macular and peripapillary retinal nerve fibre layer thickness maps and cross-sectional views demonstrating relative preservation of the inner retinal structure, including the ganglion cell complex, despite advanced outer retinal degeneration.

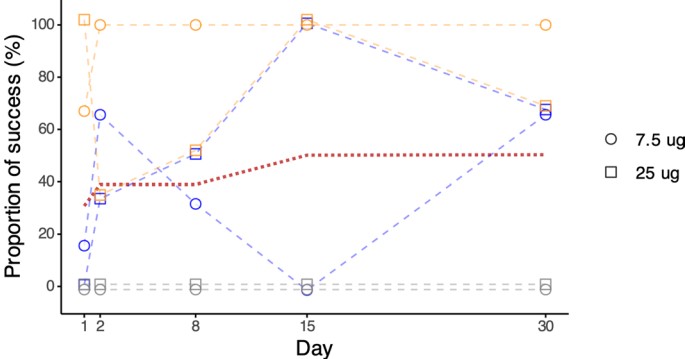

**Extended Data Fig. 2 | Individual eye-level performance on the light perception task following intravitreal KIO-301.** Data from individuals in Cohort 1 with no light perception or bare light perception visual acuity showing the proportion of correct responses on a repeated forced-choice light perception task over time following intravitreal administration of KIO-301. The task assessed the ability to detect an illuminated target presented on a dark background under monocular viewing conditions. Each line represents a single treated eye measured at an estimated photon flux of $3 \times 10^{14}$ photons $cm^{-2} s^{-1}$. Data are shown for eyes dosed at 7.5 μg or 25 μg, as indicated. Colors indicate individual participants. For visual clarity, small vertical offsets were applied to overlapping data points. A dotted red line indicates the mean trajectory across all plotted observations and is shown solely as a visual guide to aid interpretation of individual-level data; no statistical inference was performed.

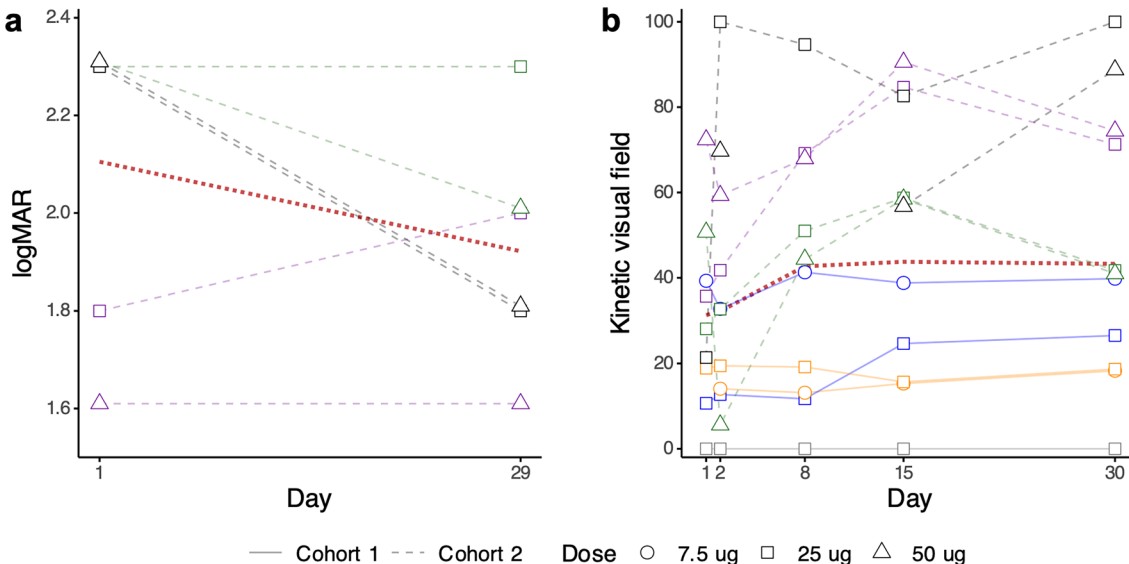

**Extended Data Fig. 3 | Visual acuity and kinetic visual field trajectories following intravitreal KIO-301 administration. (a, b)** Time courses showing individual eye-level visual acuity and kinetic visual field measurements. Day 1 represents the pre-operative (baseline) assessment obtained prior to dosing. Each symbol represents a dose, with lines connecting repeated measurements from the same eye. Colors indicate individual participants. For visual clarity, small vertical offsets were applied to overlapping data points. A dotted red line indicates the mean trajectory across all plotted observations and is shown solely as a visual guide to aid interpretation of individual-level data; no statistical inference was performed. (**a**) Measured logMAR values over time in treated eyes from participants in Cohort 2. At this level of profound visual impairment, BRVT-derived logMAR values reflect performance on simplified spatial vision tasks rather than recognition of conventional optotypes. Visual acuity was not quantifiable on this scale in Cohort 1 due to profound baseline vision loss. (**b**) Kinetic visual field scores over time in treated eyes across cohorts and dose levels. logMAR, logarithm of the minimum angle of resolution; BRVT, Berkeley Rudimentary Vision Test.

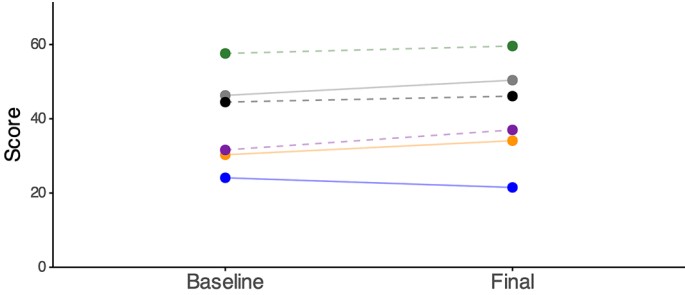

**Extended Data Fig. 4 | Individual-level changes in vision-related quality-of-life scores following KIO-301 administration.** Individual participant trajectories showing baseline (screening) and final (study completion) scores on the vision-related quality-of-life questionnaire. Each color represents a single participant, with lines connecting repeated measurements from the same participant. Data are presented descriptively to illustrate within-participant changes over time; no formal hypothesis testing was prespecified or performed.

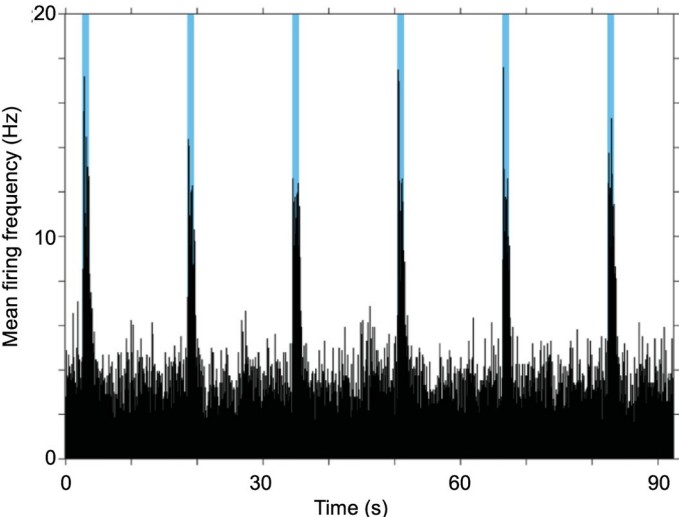

**Extended Data Fig. 5 | Preclinical validation of KIO-301 photoswitch activity in rd1 retinal explants.** Representative multielectrode array recording from an explanted rd1 mouse retina following incubation with KIO-301 (100 μM). Mean retinal ganglion cell firing frequency is shown over time, with vertical blue bars indicating periods of light stimulation (455 nm). Light-evoked spiking responses were observed during stimulus presentation, with minimal activity in darkness.

| | |
|---|---|

# Reporting Summary

## Statistics

For all statistical analyses, confirm that the following items are present in the figure legend, table legend, main text, or Methods section.

| n/a | Confirmed | |
|---|---|---|
| ☐ | ☒ | The exact sample size (*n*) for each experimental group/condition, given as a discrete number and unit of measurement |
| ☐ | ☒ | A statement on whether measurements were taken from distinct samples or whether the same sample was measured repeatedly |
| ☒ | ☐ | The statistical test(s) used AND whether they are one- or two-sided<br>*Only common tests should be described solely by name; describe more complex techniques in the Methods section.* |
| ☒ | ☐ | A description of all covariates tested |
| ☒ | ☐ | A description of any assumptions or corrections, such as tests of normality and adjustment for multiple comparisons |
| ☒ | ☐ | A full description of the statistical parameters including central tendency (e.g. means) or other basic estimates (e.g. regression coefficient) AND variation (e.g. standard deviation) or associated estimates of uncertainty (e.g. confidence intervals) |
| ☒ | ☐ | For null hypothesis testing, the test statistic (e.g. *F*, *t*, *r*) with confidence intervals, effect sizes, degrees of freedom and *P* value noted<br>*Give P values as exact values whenever suitable.* |
| ☒ | ☐ | For Bayesian analysis, information on the choice of priors and Markov chain Monte Carlo settings |
| ☒ | ☐ | For hierarchical and complex designs, identification of the appropriate level for tests and full reporting of outcomes |
| ☒ | ☐ | Estimates of effect sizes (e.g. Cohen's *d*, Pearson's *r*), indicating how they were calculated |

*Our web collection on statistics for biologists contains articles on many of the points above.*

## Software and code

Policy information about availability of computer code

| Data collection | No custom code was used for data collection. |
|---|---|
| Data analysis | No custom software or algorithms were developed for this study. Functional MRI preprocessing, visualization and descriptive assessment were performed using standard, commercially available clinical software packages, as detailed in the Supplementary Methods. |

For manuscripts utilizing custom algorithms or software that are central to the research but not yet described in published literature, software must be made available to editors and reviewers. We strongly encourage code deposition in a community repository (e.g. GitHub). See the Nature Portfolio guidelines for submitting code & software for further information.

## Data

Policy information about availability of data

All manuscripts must include a data availability statement. This statement should provide the following information, where applicable:
- Accession codes, unique identifiers, or web links for publicly available datasets
- A description of any restrictions on data availability
- For clinical datasets or third party data, please ensure that the statement adheres to our policy

De-identified participant-level data supporting the findings of this study are not publicly available due to institutional ethics restrictions, sponsor confidentiality obligations, and the small sample size of this first-in-human clinical study, which may increase re-identification risk.
The minimum dataset necessary to interpret, verify and extend the findings reported in this Article is available from the corresponding author upon reasonable

# Research involving human participants, their data, or biological material

Policy information about studies with human participants or human data. See also policy information about sex, gender (identity/presentation), and sexual orientation and race, ethnicity and racism.

| Reporting on sex and gender | Sex was recorded as a biological attribute based on participant self-report. Gender identity was not specifically assessed. The study was conducted in accordance with the SAGER (Sex and Gender Equity in Research) guidelines; however, no sex- or gender-based analyses were performed due to the small sample size and the safety-focused, first-in-human Phase 1 design. |
| --- | --- |
| Reporting on race, ethnicity, or other socially relevant groupings | Race, ethnicity, or other socially constructed or socially relevant groupings were not collected or analysed in this study. Given the small sample size and the primary focus on safety in a first-in-human Phase 1 clinical trial, analyses stratified by race or ethnicity were not performed. |
| Population characteristics | Participants were adults with advanced retinitis pigmentosa and profound vision loss, as defined by the study inclusion criteria. Relevant population characteristics, including age, sex, diagnosis, disease severity, and prior treatments, are described in the manuscript. No genotypic stratification was performed, and the study was not powered to examine associations with demographic or clinical subgroups. |
| Recruitment | Participants were recruited from specialist retinal clinics according to predefined inclusion and exclusion criteria. As a first-in-human, safety-focused Phase 1 study in a rare disease population, recruitment was necessarily limited and may be subject to selection bias toward individuals willing and eligible to participate in an early-phase interventional trial. These factors may limit generalisability of the findings. |
| Ethics oversight | The study protocol was reviewed and approved by the Central Adelaide Local Health Network Human Research Ethics Committee and registered on ClinicalTrials.gov (NCT05282953). The study was conducted in accordance with the Declaration of Helsinki and the International Conference on Harmonisation Good Clinical Practice guidelines. All participants provided written informed consent prior to participation. Full details of ethics approval are provided in the manuscript. |

Note that full information on the approval of the study protocol must also be provided in the manuscript.

# Field-specific reporting

Please select the one below that is the best fit for your research. If you are not sure, read the appropriate sections before making your selection.

☒ Life sciences  ☐ Behavioural & social sciences  ☐ Ecological, evolutionary & environmental sciences

For a reference copy of the document with all sections, see nature.com/documents/nr-reporting-summary-flat.pdf

# Life sciences study design

All studies must disclose on these points even when the disclosure is negative.

| Sample size | Twelve eyes from six participants were included. Sample size was determined by the exploratory, safety-focused design of this first-in-human Phase 1 clinical study and was not based on a formal statistical power calculation. |
| --- | --- |
| Data exclusions | No data were excluded from the analyses. |
| Replication | This was a first-in-human clinical study, and findings were not replicated in independent cohorts. |
| Randomization | Not randomized. This was an open-label, non-randomized, safety-focused Phase 1 clinical study. |
| Blinding | Not blinded. Given the open-label design and primary focus on safety, neither participants nor investigators were masked to the intervention. |

# Reporting for specific materials, systems and methods

We require information from authors about some types of materials, experimental systems and methods used in many studies. Here, indicate whether each material, system or method listed is relevant to your study. If you are not sure if a list item applies to your research, read the appropriate section before selecting a response.

## Materials & experimental systems

| n/a | Involved in the study |
|---|---|
| ☒ | Antibodies |
| ☒ | Eukaryotic cell lines |
| ☒ | Palaeontology and archaeology |
| ☐ | Animals and other organisms |
| ☐ | Clinical data |
| ☒ | Dual use research of concern |
| ☒ | Plants |

## Methods

| n/a | Involved in the study |
|---|---|
| ☒ | ChIP-seq |
| ☒ | Flow cytometry |
| ☐ | MRI-based neuroimaging |

# Animals and other research organisms

Policy information about studies involving animals; ARRIVE guidelines recommended for reporting animal research, and Sex and Gender in Research

| | |
|---|---|
| Laboratory animals | Retinal explants were obtained from rd1 mice (retinal degeneration type 1) aged postnatal day 30–60. Animals were housed under standard laboratory conditions on a 12:12 h light–dark cycle with controlled temperature (21 °C) and humidity. |
| Wild animals | This study did not involve wild animals. |
| Reporting on sex | The sex of mice was not a variable in the preclinical validation experiments and was not considered in the study design or analysis. Experiments were performed on retinal explants to confirm biological activity of the compound prior to clinical dosing. No sex-based analyses were performed. |
| Field-collected samples | This study did not involve field-collected samples. |
| Ethics oversight | All animal procedures were approved by the University of California, Berkeley Institutional Animal Care and Use Committee (protocol AUP-2016-04-8700-3) and conducted in accordance with the NIH Guide for the Care and Use of Laboratory Animals. |

Note that full information on the approval of the study protocol must also be provided in the manuscript.

# Clinical data

Policy information about clinical studies

All manuscripts should comply with the ICMJE guidelines for publication of clinical research and a completed CONSORT checklist must be included with all submissions.

| | |
|---|---|
| Clinical trial registration | This investigator-initiated, first-in-human Phase 1 study was registered at ClinicalTrials.gov (NCT05282953) prior to enrolment. |
| Study protocol | The full clinical protocol is not publicly posted owing to sponsor-confidential information related to investigational compound formulation and dosing strategy. A redacted version of the protocol sufficient to understand study design and conduct is available from the corresponding author upon reasonable request for academic purposes, subject to institutional and sponsor review and appropriate data-sharing agreements. |
| Data collection | Participants with advanced retinitis pigmentosa were recruited and followed prospectively between November 2022 and September 2023. The first participant provided consent on 3 November 2022 and the last participant provided consent on 27 March 2023. Follow-up for the final participant was completed on 8 September 2023.<br>Clinical study visits, treatment administration and safety assessments were conducted at the Royal Adelaide Hospital (Adelaide, Australia) and the Harley Eye Clinic (Adelaide, Australia). Functional vision, light perception and visual field assessments were performed in a dedicated clinical research facility in Adelaide, Australia, operated by the study sponsor. Functional MRI assessments were conducted at the South Australian Health and Medical Research Institute (SAHMRI), Adelaide, Australia. Bioanalytical sample processing was performed at Agilex Biolabs Pty Ltd (Thebarton, Australia).<br>All data were collected prospectively according to the protocol using standardized clinical, safety and exploratory functional outcome measures at scheduled study visits. |
| Outcomes | The primary objective of the study was to assess the safety and tolerability of intravitreal KIO-301 administration. Safety outcomes were pre-specified in the protocol and included the incidence and severity of adverse events, serious adverse events, and ophthalmic examination findings over the follow-up period. Secondary and exploratory outcomes included functional and imaging measures intended to assess potential biological activity and target engagement. Given the first-in-human, small-sample design, all non-safety outcomes were pre-specified as exploratory and were summarized descriptively without formal hypothesis testing. |

## Plants

| | |
|---|---|
| Seed stocks | No plant materials, seed stocks or plant-derived biological materials were used in this study. |
| Novel plant genotypes | No plant genotypes, transgenic plant lines or gene-edited plant materials were generated or used. |
| Authentication | No plant materials were used; therefore, no plant authentication procedures were required. |

## Magnetic resonance imaging

### Experimental design

| | |
|---|---|
| Design type | Task-based (visual stimulation), block design (BOLD fMRI). |
| Design specifications | Monocular visual stimulation was delivered during BOLD fMRI acquisition (treated eye stimulated; contralateral eye occluded), with stimulus presentation synchronized to image acquisition. Block-design paradigms were presented using an NNL Aktiva fMRI interface (Nordic NeuroLab). Each paradigm lasted ~5 minutes; 110 volumes were acquired for each paradigm (90 volumes for the flickering checkerboard paradigm). Testing was performed for each eye separately at baseline and repeated at 2, 14 and 28 days post-treatment; when an eye later received a higher dose, a new baseline scan was acquired ~3 months after initial treatment. |
| Behavioral performance measures | No in-scanner behavioral responses were collected (no button-press/response-time outcomes). The fMRI component assessed stimulus-evoked BOLD signal changes during passive viewing of monocular visual stimuli. |

### Acquisition

| | |
|---|---|
| Imaging type(s) | Functional MRI: task-based BOLD fMRI (T2*-weighted EPI). Structural MRI: T1-weighted 3D MPRAGE. Field mapping: dual-echo gradient-echo field maps. |
| Field strength | Clinical MRI scanner (Siemens platform). Field strength: 3 Tesla. |
| Sequence & imaging parameters | T1-weighted anatomical (3D MPRAGE): 176 sagittal slices; FOV 256 × 256 mm; slice thickness 1 mm; isotropic voxel 1 × 1 × 1 mm; TR 2300 ms; TE 2.98 ms; TI 900 ms; flip angle 9°. BOLD fMRI (gradient-echo SMS-accelerated EPI, T2*): 54 transversal slices (angled to avoid maxillary sinuses); in-plane resolution 2 × 2 mm; slice thickness 2.5 mm; FOV 192 × 192 mm; TR 3000 ms; TE 30 ms; flip angle 90°; echo spacing 0.65 ms; bandwidth 1774 Hz/Px; acceleration factor 4 (SMS 2, GRAPPA 2); 110 measurements per paradigm (90 for flickering checkerboard). Field maps (dual-echo GRE): 36 slices aligned to fMRI; slice thickness 3 mm (no gap); voxel 3 × 3 × 3 mm; FOV 192 × 192 mm; TR 400 ms; TE 4.92/7.38 ms; flip angle 60°. |
| Area of acquisition | Whole-brain acquisition with coverage including occipital visual cortex; EPI slices were acquired in a transversal orientation (angled to avoid maxillary sinuses). |
| Diffusion MRI | ☐ Used   ☒ Not used |

### Preprocessing

| | |
|---|---|
| Preprocessing software | Preprocessing used Siemens Syngo Via workstation software. Preprocessing included motion correction, spatial normalization standard template and spatial smoothing with visual inspection of artifact and signal dropout. These procedures were applied consistently across all datasets. Analyses were conducted to support exploratory evaluation of stimulus-evoked BOLD responses in this first-in-human study (Supplementary Methods).. |
| Normalization | Functional MRI data were evaluated primarily in native subject space. Where spatial normalization was applied for visualization and exploratory modelling, alignment to a standard reference template was performed using standard routines. |
| Normalization template | Standard reference template space was used for visualization and exploratory modelling where required; primary evaluation was conducted in native subject space. |
| Noise and artifact removal | Standard preprocessing steps included motion correction, masking and visual inspection for motion-related and scanner-related artifacts. Datasets were reviewed for excessive motion or signal dropout prior to analysis. |
| Volume censoring | Data were visually inspected for motion and artifacts prior to inclusion in exploratory analyses. No formal volume scrubbing procedures were applied. |

## Statistical modeling & inference

| | |
|---|---|
| Model type and settings | General linear modelling approaches were used to explore stimulus-evoked BOLD responses within individuals. Given the small sample size and first-in-human design, results are presented descriptively and were not used to support formal efficacy inference. |
| Effect(s) tested | Exploratory changes in stimulus-evoked BOLD signal over time within individuals. |

Specify type of analysis: ☐ Whole brain  ☒ ROI-based  ☐ Both

| | |
|---|---|
| Anatomical location(s) | BOLD fMRI data were acquired. Exploratory analyses focused on the occipital visual cortex. Regions of interest were defined anatomically based on structural MRI and standard neuroanatomical boundaries after normalization to standard space, with slices through the visual cortex shown for visualization of stimulus-evoked activity. |

| | |
|---|---|
| Statistic type for inference<br><br>(See Eklund et al. 2016) | Given the first-in-human design and small sample size, analyses were exploratory and descriptive, and were not used to support formal group-level voxel-wise or cluster-wise statistical inference. |
| Correction | No formal multiple-comparison–corrected group-level statistical inference is reported in the manuscript. Modelling was performed for exploratory visualization and assessment of stimulus-evoked BOLD signal changes |

## Models & analysis

| n/a | Involved in the study |
|---|---|
| ☒ | ☐ Functional and/or effective connectivity |
| ☒ | ☐ Graph analysis |
| ☒ | ☐ Multivariate modeling or predictive analysis |

nature portfolio | reporting summary

