## [Peer Review File · Nature Medicine]

Intravitreal photoswitch therapy in advanced retinitis pigmentosa: a phase 1 open-label trial

Corresponding Author: Professor Robert Casson

Version 0:

Reviewer comments:

Reviewer #1

(Remarks to the Author)

This is a phase I/II, first-in-human trial of KIO-301, a small-molecule photoswitch designed to render retinal ganglion cells light-responsive in individuals with advanced retinitis pigmentosa. Despite limitations in design and sample size, the study is potentially impactful in a population with severe visual loss and limited treatment options, and functions as a pilot study that motivates a larger randomized controlled trial rather than providing definitive efficacy evidence.

The study uses a single-arm, open-label, dose-escalation design (Figure 3) in 6 participants (12 eyes), with the primary endpoint of safety defined in Table 1 and those in Method and a broad set of secondary visual function and fMRI and quality-of-life outcomes (Figures 4-5). The dosing scheme involves unilateral dosing at a lower dose followed by contralateral dosing at a higher dose in two cohorts defined by baseline vision, which is reasonable for an exploratory ophthalmic safety study. The SAP is provided, analyses are mostly descriptive (appropriate given the small sample size and absence of prespecified hypotheses for clinical endpoints), and the CONSORT checklist appears to be followed at a high level. Despite these strengths, there are important limitations that should be more fully acknowledged.

1. Should left and right eyes of the same person (both treated) be considered as correlated sampling units? If so, several analyses appear to treat the 12 eyes as independent observations. If not, what is the rationale of pooling them as independent units for an overall mean? Also, are within-patient comparisons between right and left eyes inherently confounded by dose, time, and baseline differences?
 2. The justification that a sample size of 6 “was estimated to provide information about safety” is vague. While very small samples are common in first-in-human ocular trials, some brief explanation of how this number was chosen would improve the work.
 3. Most of the results are descriptive without inferential testing but only fMRI part reports the associated P-value (Line 228). Also, the SAP states that there were no a priori hypotheses or significance testing for clinical data, yet a formal GLM-based analysis with p-values is presented for fMRI. Given the multiple endpoints, multiple timepoints, and small sample size, I feel that the use of a single uncorrected p-value threshold for fMRI may give a misleading impression of formal statistical confirmation, which should be cautious, and I would consider it as an exploratory objective. One may emphasize effect sizes and individual responses, and “statistically significant increase” should not be over-interpreted as proof of efficacy. Similarly, the descriptions should be cautious as most of the comparisons in Figures 3-5 appear to be not “significantly” different and they may be only visually higher after KIO-301 delivery. For example, Supplement Figure 2 (only) shows changes for cohort I but not cohort II.
- In short, although descriptive analysis is appropriate for a study of this size, mixing descriptive summaries with isolated inferential tests may be confusing. Any observed changes should be interpreted cautiously as it is not to determine whether the presented changes are clinical “improvement” in the context of a nonrandomized study with small sample size and potential confounders.
4. Minor point: Figure 4a and b should be Figure 3a and b (Lines 188 and 193). There is no Figure 6 it should be Figure 5.

Reviewer #2

(Remarks to the Author)

This study proposes a photoswitch molecule KIO-301, based on previous studies in rodents, as a drug candidate for the treatment of retinitis pigmentosa in a FIH clinical trial. The authors demonstrated no drug-related adverse events after intravitreal injection of KIO-301. One out of three participants with NLP at baseline recovered light perception. Among the four tests in functional vision, temporary improvements were reported in the walking direction test, window location test and room exit tests.

While intravitreal injection of photoswitch molecule to restore vision was proven to be safe, the effectiveness is short term and minor (please see Holz FG et al, NEJM 2025). In addition, several technical issues need to be addressed.

Major comments:

1. The references associated with Fig. 1 (Neuron 2014, Scientific Reports 2017) did not provide sufficient evidences for understanding the role of P2X7 (overexpressed in rd1 mice retina and mediates the entry of KIO-301 into the RGCs). The relevant paper (Tochitsky et al Neuron 2016) should be cited for understanding how KIO-301 works. In addition, why is KIO-301 not shown in the intracellular domain of HCN in Figure 1A? What's the mechanism for KIO-301 to open HCN channel? Is there any evidence for the overexpression of P2X7 in the retina (or more specifically RGC) of RP patients?
2. In Suppl Fig. 2, the highest success rate was 50% in one patient in Cohort 1 (3 in total) and there were no observed changes in Cohort 2. In contrast, the behavioral outcome in blind mice was quite significant in Fig. 7 in Neuron 2014. Related to comment 1, is the P2X7/HCN mechanism readily applicable to human retina? Did the author check whether 5 e15 photons/cm²/s is sufficient to elicit retinal responses?
3. The improvements in visual acuity were marginal (from 2.1 to 2.0 longMAR) in Cohort2. Given the fact that there is no observed change in light perception in Cohort 2, the lighting conditions should be carefully considered. Were the participants able to see luminant subjects? In addition, why were the results for higher dose not shown?
4. It's surprising that intermediate timepoints were provided for all visual function assessments except for visual acuity tests. Explanations should be provided.
5. It is very intriguing that the kinetic visual field had an increase of 15 degrees, despite the fact that other visual tests showed modest improvements. After reading the D/BENAQ papers on animal studies, I did not find any result regarding field of vision. Is there any correlation with the results from fMRI study? What's the field of vision in fMRI?
6. For the visual function assessments, different dosages (7.5 ug, 25 ug and 50 ug) were combined except for visual acuity tests. Such dosing protocol could lead to confounding results.
7. In Figure 5, for normalization among different subjects, an ROI should be clearly defined in each time point, and BOLD change % should be calculated.
8. The effect of KIO-301 lasted for less than a month. Would a second injection be as effective as the first? In fact, in previous animal studies of D/BENAQ, all the experiments were conducted with only one injection. A second injection should be quite straightforward in mice.

Minor comments:

1. Line 188: Figure 4a should be Figure 3a.
2. Line 193: Figure 4b should be Figure 3b.
3. Pre-operative data (control) should be provided in both Fig. 3 and Fig. 4.
4. Statistical significance analysis should be conducted in both Fig. 3 and Fig. 4.
5. Line 373: photons/cm²/s should be photons/cm²/s

Reviewer #3

(Remarks to the Author)

The manuscript by Casson et al. describes a Phase I/II dose-escalation study evaluating the safety, tolerability, and preliminary efficacy of intravitreal KIO-301 in patients with retinitis pigmentosa and choroideremia. KIO-301 is a photoswitch molecule intended to incorporate into retinal ganglion cells (RGCs) and render them light-excitabile. The authors administered three different doses of KIO-301 (7.5–50 µg) across four cohorts, treating twelve eyes from six patients, and followed these patients for one month, the predefined primary-endpoint period.

Originality & Significance: This is a novel and original concept. However, the significance is limited. Based on previously published animal studies and the work presented here suggests that even in the best case, this drug will give extremely short-lived light perception to patients. Without feasibility of repeat injections, it is not clear if this drug will really help patients.

Overall, the drug appears safe and well tolerated over this one-month interval. However, there are no convincing signs of

efficacy. Most of the data presented are anecdotal and lack statistical significance and the statistical power necessary to support efficacy claims, reducing robustness of data. The reviewer strongly recommends that all results be framed exclusively as safety and tolerability signals, not evidence of efficacy. Specific comments about data, methodology, and significance are listed below:

1. The absence of any dose–response trend in the efficacy measures is concerning and suggests that the reported visual improvements—none of which reach statistical significance and all lacking placebo controls—are likely sporadic. Despite this, the manuscript makes repeated claims of efficacy, which is inappropriate and misleading.
2. Visual acuity testing is known to be highly variable. With only a single post-treatment acuity assessment, it is not acceptable to claim an improvement in visual acuity.
3. Light detection is presented as a key efficacy readout, yet the assay lacks signal-detection analysis, limiting interpretability.
4. No statistically significant changes were found in BCVA, visual field measures, or quality-of-life metrics. For example, performance in the window-location test increased from 44% to 56%, a difference that is not statistically meaningful and likely reflects normal variability in a non-validated assay. Similarly, room-exit test performance increased from 37% (13% below chance) to 60% (13% above chance). Without statistical significance—and without controlling for learning effects—these shifts should not be interpreted as evidence of treatment efficacy.
5. The only dataset showing nominal statistical significance involves fMRI pixel-intensity measurements. However, the data are presented only as bar graphs without individual patient-level data. Individual fMRI images and pixel-intensity values should be shown for each patient to allow reviewers to judge the signal. It is also unclear what the pixel intensities were compared to (e.g., baseline values, contralateral eye).
6. No retinotopic mapping of visual cortex responses is provided, which limits interpretation of the fMRI findings.
7. The manuscript lacks details regarding injection location. Assuming this was recorded for each case, these data should be included, as they may allow correlation between injection site and visual-field outcomes.

Other points:

1. Based on an EC50 of 9.5 μM from animal studies, the manuscript should explain how human dosing was derived. Did the authors develop a quantitative model of KIO-301 pharmacokinetics? This would also inform the appropriate duration of follow-up.
2. The ex vivo evaluation of drug activity is unclear. The signal shown appears markedly weaker than previously reported in animal studies, and no vehicle control is included. A dose-response assessment in the ex vivo system would be informative.
3. The text refers to fundus and OCT imaging, but no such data are presented. All recorded imaging data should be included for completeness.

Version 1:

Reviewer comments:

Reviewer #1

(Remarks to the Author)

I appreciate the authors' careful and thoughtful revisions in response to my comments. I find their responses to be responsive. The manuscript has been substantially reframed to clearly reflect the first-in-human, Phase 1 study, with all secondary outcomes now presented descriptively and without inferential claims. The authors have removed all statistical inference, acknowledged the limited sample size, and presented individual eye-level trajectories without formal analysis in order to avoid potential statistical issues, including small sample size and correlated sampling units. This makes the analysis defensible and appropriately avoids language directly implying improvement.

While this framing is appropriate and justified given the first-in-human, safety-focused Phase 1 nature of the trial, it inevitably weakens the evidentiary impact of the findings. At this stage, the merit of the work should be evaluated primarily on its scientific and translational value.

I have one remaining suggestion: the Discussion/Limitation should explicitly state that all analyses are exploratory and not intended to support efficacy inference, which may limit the strength of conclusions, but it ensures that readers interpret the findings with appropriate caution.

Reviewer #2

(Remarks to the Author)

The authors have replied to majority of my comments by toning down the conclusions, emphasizing the safety but not visual improvements in this first-in-human clinical trial. The authors also removed multiple quantitative analysis considering the small number of trials.

Following are a few remaining comments:

1. In response to Q1, the authors indicated that Figure 1 serves as a conceptual schematic. However, the gating mechanism through which KIO-301 exerts its modulatory effects on HCN channels has not yet been elucidated. In addition, there is currently no evidence for P2X7 overexpression in RP retina or RGCs. Therefore, it's not clear what specific role Figure 1 serves in the manuscript.

2. In response to Q2, the authors emphasized that the current study (focusing on safety) was not designed to establish a retinal irradiance threshold for functional responses in humans. Full-field Stimulus Test (FST) is a specialized and simple eye testing designed to measure the light sensitivity of the entire retina. Ideally, clinical reports using FST is expected for evaluating the sensitivity after the application of KIO-301.

Reviewer #3

(Remarks to the Author)

na

Response

We thank the Editor and the Reviewers for their time, expertise, and constructive evaluation of our manuscript. We greatly appreciate the thoughtful comments and suggestions, which have been invaluable in improving the clarity, rigor, and presentation of the work. We have revised the manuscript accordingly and believe that these changes have substantially strengthened the paper. Below, we provide a detailed, point-by-point response to each comment, indicating how the manuscript has been revised.

Response to Reviewer #1

We thank the reviewer for their thoughtful and constructive comments, which have helped us strengthen the clarity and framing of this Phase 1 study. We have revised the manuscript extensively to ensure that all analyses are descriptive, that no inferential interpretation is implied, and that individual-level data are clearly presented throughout. Major changes relevant to the reviewer's points include: (i) removal of all SEM values and inferential statistics; (ii) removal of reported numerical summary values and inferential statistics, with individual-level data presented throughout. Where included, mean trajectories are shown solely as visual guides to aid interpretation and do not imply statistical aggregation or independence assumptions; (iii) consistent emphasis on safety, feasibility, and exploratory pharmacodynamic observations.

We believe these changes result in a more transparent and faithful presentation of the data, with functional and imaging outcomes shown at the individual level in a manner that preserves inter-individual variability in this first-in-human Phase 1 study.

We address each comment in turn below.

Specific responses:

1. *“Should left and right eyes of the same person be considered correlated sampling units?”*

Response:

We thank the reviewer for highlighting this important point. We agree that eyes within the same

participant are not statistically independent and that within-participant right–left comparisons are confounded by baseline severity, dose sequence, and timing. The manuscript has been revised in multiple locations to remove all statistical inferences, and now presents descriptive data only, displayed in the figures at the individual level.

Revisions to the manuscript:

- a) We revised the **Statistical considerations** subsection in the Methods:

“This study was designed as a first-in-human, Phase 1 investigation with the primary objective of evaluating safety and tolerability. Accordingly, no formal hypothesis testing was prespecified, and the sample size was not determined by statistical power calculations. Rather, cohort size was selected pragmatically to enable careful safety evaluation across a limited number of participants, consistent with early-phase ophthalmic clinical studies and ethical considerations surrounding exposure to an investigational therapy.

All analyses were descriptive. Continuous and categorical outcomes are presented at the individual participant level without formal inferential testing. Functional, imaging, and participant-reported measures were evaluated to characterise variability and potential pharmacodynamic signals, and to inform the design of subsequent studies, rather than to test predefined efficacy hypotheses.”

- b) Results section (new paragraph added to the introduction of Secondary Outcomes)

“Given the first-in-human, safety-focused design and small sample size, all functional vision and neuroimaging outcomes are presented descriptively as exploratory pharmacodynamic observations. No analyses were prespecified to formally test efficacy hypotheses.”

- c) Revision of pooled mean statements to avoid implying independent inference

Several sentences previously presented pooled means in a way that could imply independent treatment of eyes. These have now been rephrased.

- Visual acuity subsection

Original:

“The mean logMAR of the participants’ treated eye in Cohort 2 improved from 2.1 +/- 0.2 at baseline to 2.0 +/- 0.1 at day 29 (mean +/- SEM, n = 3 treated eyes, 3 eyes dosed at 25 µg OD-Figure 4a) indicating a slight improvement in acuity. The mean logMAR of Cohort 2 participants treated at the higher dose was observed to have a further reduction in logMAR over the course of the study.”

Revised:

“In Cohort 2, individual eye-level trajectories are shown in **Extended Data Fig. 3a**. These values are presented descriptively in the context of known variability in visual acuity testing among individuals with ultra-low vision.”

- Kinetic visual field subsection

Original:

“Manual Goldmann kinetic perimetry with an illuminated light stimulus (440-460 nm range) was used to assess the visual fields of participants (see Methods). As shown in **Figure 4b**, the extent of the mean field of vision in the treated eye of all participants at all doses of KIO-301 increased from baseline by 15 degrees and plateaued from day 15 through day 30.”

Revised:

“Kinetic visual fields were assessed using manual Goldmann perimetry and quantified as horizontal field extent (see Methods). Individual eye-level trajectories across time are shown in **Extended Data Fig. 3b**. Changes in horizontal field extent varied substantially between eyes, with heterogeneous trajectories observed over the 30-day follow-up period.”

- d) Figures now display individual trajectories as the primary data, with any mean trajectories shown solely as visual guides.
- e) We have also added a brief integrative Results paragraph that qualitatively links functional and fMRI observations at the individual level, without introducing new data or analyses.

2. “The justification that a sample size of 6 ‘was estimated to provide information about safety’ is vague.”

Response:

We agree that additional clarification is helpful. We have revised the manuscript to explain the rationale for the sample size in the context of a first-in-human intravitreal study. The cohort size was determined pragmatically, consistent with early-phase ophthalmic trials, to allow careful safety evaluation and dose escalation while limiting exposure to a novel compound. The study was not powered for efficacy assessment, and no quantitative inference regarding functional outcomes was intended.

Revisions to the manuscript:

a) Methods: Statistical considerations

“This study was designed as a first-in-human, Phase 1 investigation with the primary objective of evaluating safety and tolerability. Accordingly, no formal hypothesis testing was prespecified, and the sample size was not determined by statistical power calculations. Rather, cohort size was selected pragmatically to enable careful safety evaluation across a limited number of participants, consistent with early-phase ophthalmic clinical studies and ethical considerations surrounding exposure to an investigational therapy.”

3. “Most of the results are descriptive... the fMRI part reports a P-value... mixing descriptive summaries with isolated inferential tests may be confusing.”

Response:

Thanks for the very constructive sanity check. We agree the inclusion of p values associated with the fMRI data was confusing and inappropriate and have revised the manuscript accordingly. All inferential statistics and p values have been removed from the fMRI analyses. The fMRI data are now presented qualitatively, using voxel-wise BOLD signal maps and are

described as exploratory pharmacodynamic observations compatible with central visual pathway activation following treatment. No formal hypothesis testing or claims of efficacy are made.

Revisions to the manuscript:

Across the manuscript, we have ensured consistent treatment of all secondary outcomes as descriptive and exploratory, with individual-level data emphasized and no statistical thresholding applied.

- a) **Results (Exploratory outcomes: functional MRI):** Removal of all p values and inferential language; reframing of findings as qualitative, exploratory pharmacodynamic observations.
- b) **fMRI figure and legend:** Updated to show representative voxel-wise BOLD maps without quantitative summaries or statistical claims.
- c) **Discussion:** Clarified that neuroimaging findings represent exploratory pharmacodynamic responses compatible with target-engagement signals rather than evidence of efficacy.
- d) **Statistical considerations:** Updated to state explicitly that no inferential analyses were performed.

4. Minor point: figure numbering errors.

Response:

We thank the reviewer for noting these errors. All incorrect figure references, including the mislabelling of Figures 4a/b and the reference to a non-existent Figure 6, have been corrected in the revised manuscript. Note that there has been some rearranging of the Figures to accommodate the editor's request to include the participant characteristics Table in the main text, and the Supplementary Figures have been re-assigned as Extended Data, in keeping with *Nature Medicine* formatting.

Response to Reviewer #2

We thank the reviewer for their careful evaluation of the mechanistic framework and functional interpretation. In response, we have revised the manuscript to improve mechanistic accuracy, ensure consistency with the preclinical literature, and further align all functional and imaging outcomes with a first-in-human, safety-led Phase 1 study. Below, we present a point-by-point response specifying the exact textual and figure-level changes made.

Reviewer comment:

“While intravitreal injection of photoswitch molecule to restore vision was proven to be safe, the effectiveness is short term and minor (please see Holz FG et al, NEJM 2025). In addition, several technical issues need to be addressed.”

Response:

We agree that the functional effects observed in this first-in-human study were transient and modest in magnitude. All references to “improved function” have been removed. The primary objective of safety was met. We acknowledge the exciting work by Holz et al. that provides an avenue for permanent visual restoration in photoreceptor-blind individuals. However, the photoswitch approach has a fundamentally different risk profile, level of invasiveness, and durability expectations. The long-term efficacy of this approach remains to be determined in further studies.

1. “The references associated with Fig. 1 (Neuron 2014, Scientific Reports 2017) did not provide sufficient evidence for understanding the role of P2X7 (overexpressed in rd1 mouse retina and mediating entry of KIO-301 into RGCs). The relevant paper (Tochitsky et al., Neuron 2016) should be cited for understanding how KIO-301 works. In addition, why is KIO-301 not shown in the intracellular domain of HCN in Fig. 1A? What is the mechanism by which KIO-301 opens HCN channels? Is there evidence for overexpression of P2X7 in the retina (or specifically RGCs) of RP patients?”

Response

Thank you for raising these points. We have revised the manuscript, Figure 1 legend, and the Discussion to clarify the proposed mechanism of action, to ensure appropriate citation of the relevant preclinical literature, and to accurately distinguish evidence derived from rodent models from what is currently known in humans.

Specific responses:

(i) *Appropriate citation supporting the proposed mechanism in Figure 1*

Response:

Thanks for suggesting the optimal citation. We agree that the mechanistic framework illustrated in Figure 1 is most directly supported by Tochitsky et al., Neuron 2016, which describes photoswitch entry into retinal ganglion cells and subsequent modulation of endogenous ion channels in rodent models of retinal degeneration.

Revisions to the manuscript:

- a) We have added an explicit citation to Tochitsky et al. at the end of the Introduction:
“The proposed mechanism of action is illustrated in Figure 1, as described by Tochitsky et al.¹¹”

- b) Revision and clarification of Figure 1: The Figure 1 legend has been substantially revised to clarify that it represents a schematic mechanistic model derived from preclinical studies in rodent models of degenerating retina, rather than a structural or human-specific mechanism.

Revised Figure 1 legend:

Fig. 1: Proposed mechanism of action of the photoswitch KIO-301.

The proposed mechanism of action has been described by Tochitsky et al.¹¹ **(a)** In retinas with advanced photoreceptor degeneration, photoswitch molecules such as KIO-301 gain access to RGCs via large-pore purinergic P2X7 receptors, which are functionally upregulated in rodent models of degenerating retina. Following cellular entry, KIO-301 is proposed to associate non-covalently with intracellular domains of hyperpolarization-activated cyclic nucleotide-gated (HCN) channels. In darkness, KIO-301 is predominantly in the trans configuration, resulting in channel block and reduced inward cation current.

(b) Upon illumination with visible light, KIO-301 undergoes reversible photoisomerisation to the cis configuration, relieving HCN channel block and permitting Na⁺ influx. This depolarizes RGCs and can trigger action potential firing. This schematic reflects a mechanistic model derived from preclinical studies of azobenzene photoswitches in degenerated retina.

(ii) *Why KIO-301 is not depicted at a defined intracellular HCN domain in Fig. 1A*

Response:

Figure 1 is intended as a conceptual schematic, not a structural model of channel–ligand interactions. While KIO-301 is proposed to act from the intracellular side following cellular entry, the precise intracellular binding site, orientation, and molecular interaction with specific HCN channel domains have not been established. Accordingly, KIO-301 is not depicted at a defined intracellular domain of HCN in Fig. 1A, to avoid implying a level of structural or molecular resolution that is not currently supported by experimental data.

(iii) *Mechanism by which KIO-301 modulates (“opens”) HCN channels*

Response:

The precise molecular gating mechanism by which KIO-301 modulates HCN channels has not been established. Preclinical studies implicate HCN channels functionally based on pharmacological inhibition experiments demonstrating loss of BENAQ-mediated photoresponsiveness following administration of selective HCN channel blockers (Tochitsky et al. Scientific Reports 2017; Neuron 2016;).^{10, 11} To provide broader biological context, we have expanded the Discussion to summarise evidence for HCN channel expression and function in mammalian and human retina.

Revisions to the manuscript (Discussion):

- a) “HCN channels contribute to retinal signalling in mammalian models,¹⁶ and human retinal single-cell transcriptomic datasets demonstrate low-level, heterogeneous expression of HCN1 and HCN4 transcripts within RGC populations.¹⁵ Consistent with this, ivabradine (a selective HCN channel inhibitor) is associated with reversible visual disturbances in humans, suggesting that modulation of retinal HCN channel activity can influence visual perception.¹⁷ Preclinical studies suggest that BENAQ-mediated responses involve HCN channels, although the underlying molecular and structural gating mechanisms remain unclear.⁸”

(iv) Evidence for P2X7 overexpression in RP retina or RGCs**Response:**

In rodent models of photoreceptor degeneration, P2X7 receptor expression is increased at the protein level, as demonstrated by immunoblotting, with immunoreactivity localising to the inner retina, including the retinal ganglion cell layer (Martínez-Gil et al.; Ye et al.).^{13,14}

In humans, available evidence is limited to baseline transcriptomic data. Single-cell RNA-sequencing of the normal adult human retina demonstrates low-level, heterogeneous P2X7 expression in subsets of retinal ganglion cells,¹⁵ but to our knowledge, there is currently no direct evidence for P2X7 overexpression in the retina, or specifically in RGCs, of patients with retinitis pigmentosa.

Revisions to the manuscript (Discussion):

- a) “In rodent models of photoreceptor degeneration, P2X7 receptor expression is increased and localises to the inner retina, including the RGC layer.^{13, 14} In humans, single-cell RNA-sequencing of the normal adult retina demonstrates low-level, heterogeneous P2X7 expression in subsets of RGCs,¹⁵ and it remains unknown whether similar degeneration-associated upregulation occurs in RP.”

References cited in response to Q2.

8. Tochitsky, I., *et al.* Restoring visual function to blind mice with a photoswitch that exploits electrophysiological remodeling of retinal ganglion cells. *Neuron* **81**, 800-813 (2014).
10. Tochitsky, I., Trautman, J., Gallerani, N., Malis, J.G. & Kramer, R.H. Restoring visual function to the blind retina with a potent, safe and long-lasting photoswitch. *Sci Rep* **7**, 45487 (2017).
11. Tochitsky, I., *et al.* How Azobenzene Photoswitches Restore Visual Responses to the Blind Retina. *Neuron* **92**, 100-113 (2016).
13. Martinez-Gil, N., *et al.* Purinergic Receptors P2X7 and P2X4 as Markers of Disease Progression in the rd10 Mouse Model of Inherited Retinal Dystrophy. *Int J Mol Sci* **23**(2022).
14. Ye, S.S., *et al.* Purinergic P2X7 receptor involves in anti-retinal photodamage effects of berberine. *Purinergic Signal* **21**, 675-685 (2025).

15. Lukowski, S.W., *et al.* A single-cell transcriptome atlas of the adult human retina. *EMBO J* **38**, e100811 (2019).
16. Kim, D., Roh, H., Lee, H.M., Kim, S.J. & Im, M. Localization of hyperpolarization-activated cyclic nucleotide-gated channels in the vertebrate retinas across species and their physiological roles. *Front Neuroanat* **18**, 1385932 (2024).
17. Borer, J.S., Fox, K., Jaillon, P., Lerebours, G. & Ivabradine Investigators, G. Antianginal and antiischemic effects of ivabradine, an I(f) inhibitor, in stable angina: a randomized, double-blind, multicentered, placebo-controlled trial. *Circulation* **107**, 817-823 (2003).

2. “In Supplementary Fig. 2, the highest success rate was 50% in one patient in Cohort 1 and there were no observed changes in Cohort 2. In contrast, the behavioural outcome in blind mice was quite significant in Fig. 7 in Neuron 2014. Related to comment 1, is the P2X7/HCN mechanism readily applicable to the human retina? Did the authors check whether 5×10^{15} photons $\text{cm}^{-2} \text{s}^{-1}$ is sufficient to elicit retinal responses?”

Response

We thank the reviewer for raising these important translational considerations. We agree that the differences between preclinical behavioural effects and the modest, variable exploratory findings in this first-in-human study require explicit contextualisation. We have revised the manuscript to address these points directly and to clarify the scope and limitations of the present study.

Specific responses:

(i) Comparison between preclinical behavioural effects and human exploratory outcomes

Response:

The robust behavioural effects reported in blind mice in preclinical studies (e.g. Tochitsky *et al.*, *Neuron* 2014) were observed under tightly controlled experimental conditions in genetically homogeneous models of advanced photoreceptor degeneration, with optimized

stimulation parameters and repeated testing paradigms. In contrast, the present study was designed as a first-in-human, Phase 1, safety-focused, dose-escalation trial, in which all functional and behavioural measures were prespecified as secondary or exploratory outcomes. The small sample size, inter-individual heterogeneity in disease severity, and conservative dosing and stimulation parameters therefore preclude direct comparison of effect magnitude between preclinical models and human participants.

Revisions to the manuscript:

- a) The **Secondary outcomes section of the Results** now opens with an explicit statement that all functional vision and neuroimaging outcomes are presented descriptively as exploratory observations, with no prespecified efficacy analyses.

“Given the first-in-human, safety-focused design and small sample size, all functional vision and neuroimaging outcomes are presented descriptively as exploratory pharmacodynamic observations. No analyses were prespecified to formally test efficacy hypotheses.”

All language implying treatment efficacy has been removed, and behavioural findings are consistently framed as exploratory throughout the Results and Discussion.

(ii) Applicability of the P2X7/HCN mechanism to the human retina

As detailed elsewhere in our response, the proposed P2X7/HCN-based mechanism is supported by **preclinical studies in rodent models of retinal degeneration**, in which photoswitch entry via purinergic P2X receptors and functional involvement of HCN channels have been demonstrated (Tochitsky *et al.*, *Neuron* 2014, 2016).

Revisions to the manuscript

- a) In the revised **Discussion**, we now explicitly distinguish this preclinical evidence from what is currently known in humans. Specifically, we state that:

“In rodent models of photoreceptor degeneration, P2X7 receptor expression is increased and localises to the inner retina, including the RGC layer.^{13, 14} In humans, single-cell RNA-sequencing of the normal adult retina demonstrates low-level, heterogeneous P2RX7

expression in subsets of RGCs,¹⁵ and it remains unknown whether similar degeneration-associated upregulation occurs in RP.”

- b) Explicit acknowledgement of translational uncertainty included in the extensively expanded **Limitations** subsection of the Discussion:

“This study has several important limitations. The sample size was necessarily small, consistent with a first-in-human, Phase 1 dose-escalation design, limiting the ability to draw quantitative or generalisable conclusions. The open-label, single-arm nature of the trial and the absence of a control group preclude causal inference and do not allow efficacy to be assessed. Multiple functional, imaging, and participant-reported outcomes were explored without formal adjustment for multiplicity, and these measures should therefore be interpreted as exploratory and hypothesis-generating rather than confirmatory. The duration of follow-up was short relative to the anticipated pharmacodynamic time course of photoswitch activity and was not designed to assess durability of effect. In addition, while the light intensity used in this study was selected on the basis of preclinical data and conservative ocular safety considerations, the relationship between photon flux, retinal engagement, and behavioural response in humans remains uncertain and was not optimised in this first-in-human, safety-focused study. Finally, while preclinical studies provide a biologically plausible framework for photoswitch-mediated RGC photosensitisation, direct mechanistic confirmation in human retinal tissue was not feasible in this study.”

(iii) Light intensity and photon flux

Response:

In the initial mechanistic studies using first-generation photoswitches (e.g. DENAQ), retinal responses were observed over several orders of magnitude, from approximately 10^{13} to 10^{16} photons $\text{cm}^{-2} \text{s}^{-1}$, with more consistent and robust physiological and behavioural responses occurring at higher intensities (Tochitsky et al., *Neuron* 2014). Subsequent studies using the more potent second-generation photoswitch BENAQ typically employed a single, representative photon flux within this effective range (approximately 2.5×10^{15} photons $\text{cm}^{-2} \text{s}^{-1}$, as the Reviewer quite rightly notes) to demonstrate reliable retinal and behavioural responses in blind mice, rather than systematically mapping response thresholds (Tochitsky et al., *Sci Rep* 2017).

The illumination levels used in the present clinical study were selected toward the lower end of this preclinical photoresponse range and were specified conservatively for safety rather than efficacy optimisation. As a result, the study was not designed to establish a retinal irradiance threshold for functional responses in humans, and the exploratory behavioural findings should be interpreted in the context of a safety-led first-in-human investigation.

As clarified in the revised Methods, preclinical data informed a conservative, safety-led dose-escalation strategy rather than targeting a putative efficacy threshold. The exploratory functional assessments reported in **Supplementary Fig. 2 (now Extended Data Fig. 2)** should therefore be interpreted in the context of a Phase 1 safety study that was not designed or powered to replicate the magnitude of behavioural effects observed in preclinical models.

Revisions to the manuscript:

- a) In the Methods subsection: added as a final paragraph to **Influence of preclinical data on trial design:**

“Preclinical studies demonstrated photoswitch-mediated retinal responses across a broad range of photon fluxes under controlled experimental conditions in which retinal illumination could be estimated directly. In contrast, retinal photon flux cannot be directly measured in vivo in human participants; therefore, light stimulation parameters in this first-in-human study were primarily selected conservatively based on incident illumination and ocular safety considerations and corresponded with preclinical levels towards the lower end of photoresponsiveness.”

- b) The **Limitations** section in the **Discussion** has been expanded and now includes:

“In addition, while the light intensity used in this study was selected on the basis of preclinical data and conservative ocular safety considerations, the relationship between photon flux, retinal engagement, and behavioural response in humans remains uncertain and was not optimised in this first-in-human, safety-focused study.”

3. “The improvements in visual acuity were marginal (from 2.1 to 2.0 logMAR) in Cohort 2.

Given the fact that there is no observed change in light perception in Cohort 2, the lighting conditions should be carefully considered. Were the participants able to see luminant subjects? In addition, why were the results for higher dose not shown?”

Response

We thank the reviewer for this comment and agree that consideration of the visual acuity methodology and lighting conditions is important. Visual acuity in this study was assessed using the Berkeley Rudimentary Vision Test (BRVT), which is specifically designed to quantify residual visual function in individuals with profound vision loss and does not rely on conventional optotype recognition.

BRVT testing was conducted under standardized conditions, and performance reflects the ability to detect or localize high-contrast stimuli rather than fine spatial resolution. Accordingly, the modest changes observed in Cohort 2 reflect limited improvements in task performance under these defined testing conditions.

Participants in Cohort 2 could see luminous objects and retained sufficient residual vision to permit BRVT-based measurement, whereas those in Cohort 1 had no light perception or only bare light perception and therefore could not generate meaningful BRVT scores. For this reason, only Cohort 2 is shown in the visual acuity figure. The data for the 25 μg and 50 μg doses are shown in the revised figure (now Extended Data Fig. 3a).

Importantly, the present study was not designed or powered to detect efficacy signals, and visual acuity outcomes were included as exploratory measures. The absence of consistent changes in forced-choice light perception tasks does not preclude limited improvements in spatial perception under specific stimulus conditions, nor does it imply inconsistency within the dataset. Rather, these findings reflect the expected variability and constraints of early-phase, safety-focused evaluation.

Revisions to the manuscript

- a) Visual acuity figure (now Extended Data Fig. 3a) has been revised to show individual eye-level data with a mean trend shown as a visual aid only without inferential statistics. All administered dose levels shown where testing was feasible
- b) Added to the Results (Visual acuity subsection) and the figure legend:
“At this level of profound visual impairment, BRVT-derived logMAR values reflect performance on simplified spatial vision tasks rather than recognition of conventional optotypes.”

4. *“It’s surprising that intermediate timepoints were provided for all visual function assessments except for visual acuity tests. Explanations should be provided.”*

Response

We thank the reviewer for this observation and agree that clarification is warranted.

In this study of individuals with ultra-low vision, the protocol prioritised functional visual assessments over acuity-based measures. Quantitative visual acuity testing using the Berkeley Rudimentary Vision Test (BRVT) was introduced as a protocol amendment after study initiation and was specified only at baseline and Day 30. As a result, visual acuity was not assessed at intermediate visits.

By contrast, the other visual function assessments were prespecified in the original protocol and therefore performed at multiple timepoints. The absence of intermediate visual acuity measurements thus reflects protocol-defined assessment timing rather than missing data or selective reporting.

Revisions to manuscript:

a) This has now been clarified in the Results section:

“Visual acuity assessment using the Berkeley Rudimentary Vision Test was introduced as a protocol amendment and was therefore performed at baseline and Day 29-30 only.”

5. *“It is very intriguing that the kinetic visual field had an increase of 15 degrees, despite the fact that other visual tests showed modest improvements. After reading the D/BENAQ papers on animal studies, I did not find any result regarding field of vision. Is there any correlation with the results from fMRI study? What’s the field of vision in fMRI?”*

Response

We agree that the apparent change in kinetic visual field extent should be interpreted cautiously in the context of a first-in-human, safety-focused study with ultra-small sample size and exploratory functional outcomes.

In ABACUS-1, fMRI was included as an exploratory measure of stimulus-evoked cortical activity rather than as a retinotopic or perimetric mapping tool. The visual stimuli used in this study extended to approximately 25 degrees of visual angle, which is typical of standard fMRI visual stimulation paradigms. However, these paradigms are not designed to define a

behavioural field of vision and cannot be directly compared with degrees of visual field extent measured by Goldmann kinetic perimetry. Accordingly, we did not attempt to correlate fMRI findings with kinetic visual field changes in this Phase 1 study.

6. “For the visual function assessments, different dosages (7.5 µg, 25 µg and 50 µg) were combined except for visual acuity tests. Such dosing protocol could lead to confounding results.”

Response

We thank the reviewer for this comment and agree that pooling across dose levels requires careful interpretation.

In the original submission, exploratory visual function outcomes were summarised using group means and measures of dispersion. In response to reviewer feedback, **all visual function figures have now been revised to present individual eye-level data only**, with removal of mean curves and SEM. This revision was made specifically to avoid any implication of group-level effects or dose–response relationships in this first-in-human, safety-focused study.

The study was not designed or powered to assess efficacy or dose–response. Accordingly, **visual function assessments are presented descriptively**, to illustrate feasibility of testing and inter-individual variability rather than to support comparative conclusions between dose levels. Pooling across dose cohorts in these descriptive displays does not imply equivalence or infer dose-dependent effects.

Visual acuity outcomes were analysed separately for methodological reasons related to test suitability, not dose selection, as clarified in the revised Results and Extended Data Figure 3 legend. No conclusions regarding comparative efficacy or dose–response are drawn from any visual function assessments.

7. “In Figure 5, for normalization among different subjects, an ROI should be clearly defined in each time point, and BOLD change % should be calculated.”

Response

We thank the reviewer for this comment and agree that quantitative comparison of BOLD signal across subjects would require a clearly defined region of interest (ROI) and calculation of percent BOLD signal change.

In the original submission, Figure 5 included exploratory quantitative summaries of fMRI activation. In response to reviewer feedback and to avoid overinterpretation of ultra-small-n neuroimaging data, **all quantitative fMRI analyses have now been removed**. The revised figure (now Fig. 4) presents **qualitative fMRI images only**, intended to illustrate stimulus-evoked cortical signal patterns at the individual level rather than to support quantitative or comparative inference.

Accordingly, no ROI-based normalization or percent BOLD change calculations are now presented, and no conclusions regarding magnitude, spatial extent, or inter-subject comparability of fMRI responses are drawn. We present an inset showing the V1 and V2 regions. This change is consistent with the safety- and feasibility-focused aims of this first-in-human Phase 1 study and with the revised framing of fMRI as an exploratory, descriptive measure of potential target engagement only.

8. “The effect of KIO-301 lasted for less than a month. Would a second injection be as effective as the first? In fact, in previous animal studies of D/BENAQ, all the experiments were conducted with only one injection. A second injection should be quite straightforward in mice.”

Response

We thank the reviewer for these thoughts. The present study was designed as a first-in-human, single-administration Phase 1 trial, with the primary objective of assessing safety and feasibility. Accordingly, only a single intravitreal injection of KIO-301 was administered, and the study was not designed to evaluate the effects, durability, or repeatability of repeated dosing.

Preclinical studies of azobenzene photoswitches have likewise focused on single-administration paradigms to establish proof-of-concept and mechanism of action. Consideration of repeat dosing, including durability and dosing frequency, was outside the scope of this initial clinical investigation. It is nevertheless likely that, if this therapeutic approach were to be developed further, clinical treatment would require repeated intravitreal administration at regular intervals, analogous to some other intravitreally delivered therapies.

Minor points:

1. Line 188: Figure 4a should be Figure 3a.

2. Line 193: Figure 4b should be Figure 3b.

Thanks for finding these errors. Note that Fig. 3 is now Extended Data Fig. 3

3. Pre-operative data (control) should be provided in both Fig. 3 and Fig. 4.

We thank the reviewer for this suggestion. Pre-operative measurements were obtained for all participants prior to dosing and are included as the Day 1 timepoint in the figures. The figure legends have been revised to explicitly state that Day 1 represents the pre-operative (baseline) assessment.

4. Statistical significance analysis should be conducted in both Fig. 3 and Fig. 4.

In responding to the editor and other Reviewers, the figures have been modified to show trajectories at the individual level with no statistical inferences (the mean trajectory is shown as a visual aid only).

5. Line 373: photons/cm²/s should be photons/cm²/s

Thanks for finding these errors. Now corrected, noting that the main Methods have been streamlined and this technical detail has been moved to the Supplementary Material.

Response to Reviewer #3

General

We thank the reviewer for their careful and thoughtful assessment of this first-in-human study. We acknowledge that ABACUS-1 was designed to evaluate **safety, tolerability, and feasibility**, and was **not intended to establish clinical efficacy**. In response to reviewer and editorial guidance, the manuscript has been extensively revised to remove all claims of efficacy, to present all functional and imaging outcomes descriptively at the individual level, and to frame these findings explicitly as **exploratory only**. No inferential statistical analyses are presented, no dose–response conclusions are drawn, and the absence of a control group, small sample size, and short follow-up are explicitly acknowledged as major limitations.

Specific responses:

Originality & Significance:

“This is a novel and original concept. However, the significance is limited... even in the best case, this drug will give extremely short-lived light perception... Without feasibility of repeat injections, it is not clear if this drug will really help patients.”

Response

We agree that, based on preclinical and early human data, any photoswitch-mediated effects are likely to be transient, and that this study does not address the durability or repeatability of treatment. ABACUS-1 was designed as a first-in-human, single-administration Phase 1 study, with the primary objective of establishing ocular and systemic safety rather than long-term functional benefit.

We respectfully note, however, that the clinical significance of this study lies in feasibility and safety, rather than durability of effect. Demonstrating that a small-molecule photoswitch can be delivered intravitreally in humans without inflammation, structural retinal toxicity, or systemic exposure, and with signals consistent with pharmacodynamic target engagement, is a necessary prerequisite for any subsequent evaluation of repeat dosing, optimisation of light stimulation, or assessment of sustained benefit.

The feasibility of repeat intravitreal administration was not evaluated in this study by design, but is supported indirectly by the favourable safety profile observed and by the extensive clinical precedent for repeated intravitreal injections in retinal disease. Whether repeated dosing would be effective, tolerable, or clinically meaningful remains an open question and is appropriately reserved for future studies.

Revisions to manuscript

- b) The manuscript now avoids claims regarding long-term benefit and frames all functional observations as exploratory, intended to inform subsequent trial design rather than to establish therapeutic efficacy.

Sentence added to the conclusion of the revised Discussion.

- c) “Consistent with preclinical data, any photoswitch-mediated effects observed were transient, and the study was not designed to assess durability, repeat dosing, or clinical efficacy.”

“Overall, the drug appears safe and well tolerated over this one-month interval. However, there are no convincing signs of efficacy. Most of the data presented are anecdotal and lack statistical significance and the statistical power necessary to support efficacy claims, reducing robustness of data. The reviewer strongly recommends that all results be framed exclusively as safety and tolerability signals, not evidence of efficacy.”

Response

We thank the reviewer for this helpful sanity check. We agree entirely with the assessment. The manuscript has been comprehensively revised to frame all findings **exclusively** in terms of safety, tolerability, feasibility, and exploratory pharmacodynamic observations, and **not as evidence of clinical efficacy**.

- All language implying efficacy or vision restoration has been removed.
- No inferential statistical analyses are presented.
- All functional, imaging, and participant-reported outcomes are described as descriptive, exploratory, and hypothesis-generating only, with individual-level data shown throughout.
- Figures have been reconstructed to present descriptive or qualitative data emphasized at the individual level.
- The absence of statistical power, the lack of a control group, and the limited duration of follow-up are explicitly acknowledged as major limitations.
- Accordingly, no conclusions regarding efficacy are drawn in the revised manuscript.

1. “The absence of any dose–response trend in the efficacy measures is concerning and suggests that the reported visual improvements—none of which reach statistical significance and all lacking placebo controls—are likely sporadic. Despite this, the manuscript makes repeated claims of efficacy, which is inappropriate and misleading.”

Response

The manuscript has been revised extensively to remove all claims of efficacy and to avoid any implication of dose–response relationships.

- No dose–response trend is claimed or inferred in the revised manuscript. All language describing “improvement,” “benefit,” or “efficacy” has been removed or replaced with neutral, descriptive terminology.

- All functional and imaging outcomes are now presented at the individual level only, without inferential statistics or group-level comparisons.
- The absence of placebo controls, limited statistical power, and inter-individual variability are explicitly acknowledged as major limitations.
- Accordingly, all visual and functional findings are framed as exploratory and hypothesis-generating, and not as evidence of treatment efficacy.

Revisions to the manuscript:

- a) Title; Abstract; Results (Secondary and Exploratory Outcomes); Discussion (including Limitations).

2. “Visual acuity testing is known to be highly variable. With only a single post-treatment acuity assessment, it is not acceptable to claim an improvement in visual acuity.”

Response

We agree with the reviewer that visual acuity testing at this level of profound visual impairment is highly variable and that a single post-treatment assessment does not support claims of improvement. In response, all language implying improvement in visual acuity has been removed, and visual acuity outcomes are now presented descriptively only.

- Visual acuity is now explicitly described as a variable measure in individuals with ultra-low vision.
- No claims of visual acuity improvement are made in the revised manuscript. Visual acuity values are presented as individual eye-level data, without inferential statistics (the mean trend is shown in Extended Fig. 3 as a visual aid).
- The limited timing of post-treatment visual acuity assessment is explicitly acknowledged as a methodological constraint.

Revisions to the manuscript:

- a) Results (Visual acuity subsection) now reads:

“In Cohort 2, individual eye-level trajectories are shown in **Extended Fig. 3a**. These values are presented descriptively in the context of known variability in visual acuity testing among individuals with ultra-low vision.”

b) Corresponding figure legend now reads:

“Extended Fig. 3: Visual acuity and kinetic visual field trajectories following intravitreal KIO-301 administration.

a–b, Time courses showing individual eye-level visual acuity and kinetic visual field measurements. Day 1 represents the pre-operative (baseline) assessment obtained prior to dosing. Each symbol represents a dose, with lines connecting repeated measurements from the same eye. Colors indicate individual participants. For visual clarity, small vertical offsets were applied to overlapping data points. **(a)** Measured logMAR values over time in treated eyes from participants in Cohort 2. At this level of profound visual impairment, BRVT-derived logMAR values reflect performance on simplified spatial vision tasks rather than recognition of conventional optotypes. Visual acuity was not quantifiable on this scale in Cohort 1 due to profound baseline vision loss.”

3. “Light detection is presented as a key efficacy readout, yet the assay lacks signal-detection analysis, limiting interpretability.”

We agree with the reviewer that the light-detection task lacks formal signal-detection analysis and therefore has limited interpretability. In response, light-detection outcomes are no longer presented as an efficacy readout and are framed explicitly as exploratory, descriptive observations.

- The light-detection task is no longer described or interpreted as evidence of treatment efficacy.
No signal-detection or inferential statistical analyses are presented.
- Light-detection data are presented descriptively at the individual eye level only. The methodological limitations of this assay are explicitly acknowledged in the Discussion.

Revisions to the manuscript:

Results (Light perception subsection); Extended Data Fig. 2 and legend have been modified:

- a) “Light perception was assessed using a repeated forced-choice task. Individual eye-level data are shown in **Extended Data Fig. 2**. In Cohort 1, which comprised participants who were NLP or BLP at baseline, variability in task performance was observed across study visits, with some participants demonstrating non-zero responses at post-treatment assessments.”
- b) **“Extended Data Fig. 2: Individual eye-level performance on the light perception task following intravitreal KIO-301.**

Data from individuals in Cohort 1 with no light perception or bare light perception visual acuity showing the proportion of correct responses on a repeated forced-choice light perception task over time following intravitreal administration of KIO-301. The task assessed the ability to detect an illuminated target presented on a dark background under monocular viewing conditions. Each line represents a single treated eye measured at an estimated photon flux of 3×10^{14} photons·cm⁻²·s⁻¹. Data are shown for eyes dosed at 7.5 µg or 25 µg, as indicated. Colors indicate individual participants. For visual clarity, small vertical offsets were applied to overlapping data points.”

4. “No statistically significant changes were found in BCVA, visual field measures, or quality-of-life metrics. For example, performance in the window-location test increased from 44% to 56%, a difference that is not statistically meaningful and likely reflects normal variability in a non-validated assay. Similarly, room-exit test performance increased from 37% (13% below chance) to 60% (13% above chance). Without statistical significance—and without controlling for learning effects—these shifts should not be interpreted as evidence of treatment efficacy.”

Response

We agree with the reviewer that the observed changes in behavioural and functional vision tasks are modest, variable, and lack statistical significance, and that, without controlling for learning effects, they should not be interpreted as evidence of treatment efficacy. In response, all such outcomes are now framed explicitly as **exploratory and descriptive**, without inferential interpretation.

- No statistically significant changes are reported for BCVA, kinetic visual fields, functional vision tasks, or quality-of-life measures.
- No behavioural or functional outcomes are interpreted as evidence of efficacy. Potential learning effects and task variability are explicitly acknowledged as important limitations.
All behavioural and functional vision data are presented at the individual level only, without inferential statistics or comparison to chance performance.
- The absence of statistical power, validated efficacy endpoints, and control conditions is clearly stated.

Revisions to the manuscript:

- a) The opening paragraph of the Results now reads:

“Given the first-in-human, safety-focused design and small sample size, all functional vision and neuroimaging outcomes are presented descriptively as exploratory pharmacodynamic observations. No analyses were prespecified to formally test efficacy hypotheses.”

- b) Figures and legends have been modified to show individual level data only. No statistical inferences are drawn.

- c) An extensive limitations section is now in the Discussion and includes:

“The sample size was necessarily small, consistent with a first-in-human, Phase 1 dose-escalation design, limiting the ability to draw quantitative or generalizable conclusions. The open-label, single-arm nature of the trial and the absence of a control group preclude causal inference and do not allow efficacy to be assessed. Multiple functional, imaging, and participant-reported outcomes were explored without formal adjustment for multiplicity, and these measures should therefore be interpreted as exploratory and hypothesis-generating rather than confirmatory.”

5. *“The only dataset showing nominal statistical significance involves fMRI pixel-intensity measurements. However, the data are presented only as bar graphs without individual patient-level data. Individual fMRI images and pixel-intensity values should be shown for each patient to allow reviewers to judge the signal. It is also unclear what the pixel intensities were compared to (e.g., baseline values, contralateral eye).”*

Response

We agree with the reviewer that presentation of fMRI data as aggregated bar graphs with nominal statistical significance was inappropriate in the context of this ultra-small, first-in-human Phase 1 study. In response, the fMRI analyses and figures have been comprehensively revised.

- All quantitative fMRI summaries, bar graphs, pixel-intensity comparisons, and associated p values have been removed.
- fMRI data are now presented as **individual-level, qualitative BOLD signal maps** for each participant, allowing direct visual assessment of task-associated cortical signal. fMRI images are displayed **relative to baseline** under identical stimulus conditions, which is now explicitly stated in the figure legend and Results text.
- No comparisons with the contralateral eye or group-level intensity metrics are performed or implied.
- fMRI findings are now framed exclusively as **exploratory, descriptive observations consistent with pharmacodynamic target engagement**, and not as evidence of efficacy.

Revisions to the manuscript:

- a) Results (Exploratory outcomes: functional MRI); fMRI and legend; Statistical considerations; Discussion.

6. *“No retinotopic mapping of visual cortex responses is provided, which limits interpretation of the fMRI findings.”*

Response

We agree with the reviewer that the absence of retinotopic mapping limits interpretation of the fMRI findings. Retinotopic mapping was not performed in this study, as fMRI was included

solely as an **exploratory measure of stimulus-evoked cortical activity** rather than as a tool for spatial or functional visual field mapping.

- fMRI was not designed or intended to provide retinotopic localisation or quantitative structure–function correlation.
- All fMRI findings are now presented and discussed as **qualitative, exploratory observations** only. This limitation is explicitly acknowledged in the Discussion.

Revisions to the manuscript:

- a) Results (Exploratory outcomes: functional MRI); Discussion (Limitations).

The fMRI results section now reads:

“Qualitative fMRI blood-oxygen-level-dependent (BOLD) signal maps showed suprathreshold task-associated signal following KIO-301 administration (**Fig. 4**). Across participants, signal was observed in multiple cortical regions, including within occipital cortex encompassing primary visual cortex (V1) and adjacent extrastriate areas, at early post-treatment time points. Visually evoked cortical signal was most prominent within 2–3 days following treatment and showed reduced spatial extent at later time points. These observations are exploratory and descriptive and were not designed to establish visual perception or treatment efficacy.”

7. “The manuscript lacks details regarding injection location. Assuming this was recorded for each case, these data should be included, as they may allow correlation between injection site and visual-field outcomes.”

Response

We thank the reviewer for this suggestion. Injection location was not recorded in a manner that would allow formal analysis. All intravitreal injections were performed by experienced eye surgeons, including R.J.C., using standard clinical practice, which in all cases involved injection in the superotemporal quadrant.

- Injection site was therefore not systematically varied across participants.

- No study procedures were designed to assess structure–function relationships based on injection location.
- Given the uniform injection approach and the exploratory nature of functional outcomes in this Phase 1 study, correlation between injection site and visual-field outcomes would not be informative.
- This limitation is now acknowledged implicitly through the revised framing of all functional outcomes as exploratory and descriptive only.

Other points:

1. “Based on an EC50 of 9.5 μ M from animal studies, the manuscript should explain how human dosing was derived. Did the authors develop a quantitative model of KIO-301 pharmacokinetics? This would also inform the appropriate duration of follow-up.”

Response

We agree that human dose selection in a first-in-human study should not be based on extrapolation from an EC50 alone. Assuming uniform vitreous distribution, the highest dose would correspond to a low-tens-of-micromolar concentration, within the range associated with biological activity in preclinical models; however, **dose selection was safety-led** rather than EC50-driven

The manuscript has been revised to explain how preclinical retinal explant and in vivo data informed a conservative starting dose and escalation approach, while prioritising safety and feasibility as the primary objectives.

The in vitro EC50 (9.5 μ M; 4.5 μ g/mL) from rd1 retinal explants is acknowledged as a pharmacology benchmark but was **not used as the basis for human dose selection**.

A quantitative human PK/PD model was **not developed** for this Phase 1 safety study. Preclinical PK and pharmacodynamic time course data were used to inform the **timing of early post-dose assessments**, consistent with a short-acting pharmacodynamic profile described in the revised Methods

Preclinical ocular PK studies in pigmented rabbits report a terminal half-life in retina of **326 hours (14 days)** after IVT dosing with the clinical formulation, supporting the study's assessment schedule and the emphasis on early post-dose evaluation.

ABACUS-1 follow-up duration and escalation decisions were predefined and driven by safety considerations; escalation was contingent on absence of dose-limiting toxicities and was **not based on functional outcomes**.

Revisions to the manuscript:

a) **Methods (Influence of preclinical data on trial design)**

Clarifies that preclinical EC50 values informed biological plausibility but **did not drive human dose selection**, which was safety-led:

“These findings informed the selection of a conservative starting dose for first-in-human evaluation and supported a safety-led dose-escalation strategy rather than targeting a putative efficacy threshold.”

b) **Methods (Influence of preclinical data on trial design)**

Explains how preclinical pharmacodynamics informed the **timing of assessments**, not efficacy modelling:

“Preclinical studies also indicated that photoswitch-mediated responses occur rapidly following intraocular administration and decline over days to weeks, consistent with a short-acting pharmacodynamic profile. Accordingly, the clinical study design incorporated early post-dosing assessments...”

c) **Methods (Intervention and dose escalation)**

States explicitly that dose escalation was **based on safety criteria only**, not functional outcomes or PK/PD modelling:

“Advancement to higher dose levels was contingent on the absence of dose-limiting ocular or systemic adverse events and was based solely on predefined safety criteria, not on functional or imaging outcomes.”

d) **Methods (Statistical considerations)**

Clarifies that the study was **not designed or powered for efficacy or PK/PD modelling**:

“This study was designed as a first-in-human, Phase 1 investigation with the primary objective of evaluating safety and tolerability. Accordingly, no formal hypothesis testing was prespecified, and the sample size was not determined by statistical power calculations. Rather, cohort size was selected pragmatically to enable careful safety evaluation across a limited number of participants, consistent with early-phase ophthalmic clinical studies and ethical considerations surrounding exposure to an investigational therapy.

All analyses were descriptive. Continuous and categorical outcomes are presented at the individual participant level without formal inferential testing. Functional, imaging, and participant-reported measures were evaluated to characterise variability and potential pharmacodynamic signals, and to inform the design of subsequent studies, rather than to test predefined efficacy hypotheses.”

Discussion (Limitations)

Explicitly acknowledges that **durability, repeat dosing, and formal PK/PD relationships** were outside the scope of this Phase 1 study:

“The duration of follow-up was short relative to the anticipated pharmacodynamic time course of photoswitch activity and was not designed to assess durability of effect.”

2. “The ex vivo evaluation of drug activity is unclear. The signal shown appears markedly weaker than previously reported in animal studies, and no vehicle control is included. A dose-response assessment in the ex vivo system would be informative.”

Response

We agree that the purpose and limitations of the ex vivo retinal explant experiments should be clearly articulated. The ex vivo studies included in this manuscript were designed to **confirm biological activity of the clinical formulation of KIO-301 prior to human dosing**, rather than to replicate or optimise pharmacological dose–response relationships previously established in animal models.

- The ex vivo experiments were performed using a **single, supra-threshold concentration** selected to demonstrate light-evoked retinal responses under the specific experimental conditions employed, consistent with their role as qualitative confirmation of activity rather than pharmacological characterisation.
- The magnitude of the ex vivo signal differs from that reported in some prior animal studies, which likely reflects differences in species, retinal preparation, disease model, recording methodology, and stimulation paradigm, and does not imply reduced biological activity.
- A formal multi-point **dose–response analysis was not performed** in the ex vivo system, as this was outside the scope of the study and not required to support the safety-led objectives of this first-in-human clinical trial.
- We agree that dose–response assessment and direct comparison with prior animal pharmacology would be informative; however, these questions are more appropriately addressed in dedicated preclinical optimisation studies rather than within the context of a Phase 1 clinical investigation.
- The ex vivo data are now explicitly described as **qualitative and supportive**, and no claims regarding potency, efficacy, or translational equivalence to animal studies are made.

Revisions to the manuscript:

- a) Added to Methods (Preclinical retinal explant validation of photoswitch activity)

“These experiments were intended as qualitative confirmation of biological activity of the clinical formulation and were not designed to assess dose–response relationships or comparative potency relative to prior animal studies.”

- b) **Methods (Influence of preclinical data on trial design)**

Frames all preclinical experiments as supportive of biological plausibility and safety-led clinical translation, rather than efficacy or pharmacological optimisation.

c) **Discussion (Limitations)**

Explicitly states that optimisation of dosing, formal pharmacodynamic characterisation, and efficacy assessment were outside the scope of this first-in-human Phase 1 study.

3. “The text refers to fundus and OCT imaging, but no such data are presented. All recorded imaging data should be included for completeness.”

Response

We thank the reviewer for this comment. Retinal imaging was performed as part of protocol-defined ocular safety monitoring and included colour fundus photography, fundus autofluorescence (AF), and optical coherence tomography (OCT) of the macula and peripapillary retinal nerve fibre layer (RNFL). **These assessments demonstrated findings typical of advanced retinitis pigmentosa and did not reveal treatment-related structural retinal abnormalities following intravitreal administration of KIO-301.** Imaging assessments were conducted for safety monitoring and were not intended as efficacy endpoints. To address the reviewer’s request for completeness, **representative retinal imaging is now provided in Extended Data Fig. 1**, including colour fundus photography, AF, macular OCT, and RNFL OCT. The Results and Methods sections have been updated accordingly to reflect these assessments.

Revisions to the manuscript

a) **Results (Primary outcome: Safety and tolerability)**

Now references Extended Data Fig. 1 and summarises structural retinal imaging findings as part of ocular safety monitoring.

b) **Methods (Safety assessments)**

Clarified to include colour fundus photography, fundus autofluorescence, and OCT (macula and RNFL) as protocol-defined safety assessments.

c) **Extended Data (Fig. 1)**

Added representative retinal imaging demonstrating typical advanced retinitis pigmentosa morphology and absence of treatment-related structural abnormalities, with relative preservation of the inner retina.

Point-by-point response to reviewers

We thank the reviewers for their careful reading of the revised manuscript and for their constructive final comments. We have addressed each point below and have made corresponding clarifications in the manuscript.

Reviewer #1

Comment:

“I have one remaining suggestion: the Discussion/Limitation should explicitly state that all analyses are exploratory and not intended to support efficacy inference, which may limit the strength of conclusions, but it ensures that readers interpret the findings with appropriate caution.”

Response:

We thank the reviewer for this helpful suggestion and agree that explicit clarification is important. The manuscript has been revised to state clearly that all functional, imaging and participant-reported outcomes are exploratory and descriptive and were not designed or powered to support efficacy inference. This clarification has been added to the Discussion/Limitations section to ensure appropriate interpretation of the findings.

Change made in manuscript:

“Accordingly, all functional, imaging and participant-reported outcomes in this study are exploratory and descriptive and were not designed or powered to support efficacy inference.”

Reviewer #2

We thank the reviewer for their careful evaluation and constructive final comments.

Comment 1

“In response to Q1, the authors indicated that Figure 1 serves as a conceptual schematic. However, the gating mechanism through which KIO-301 exerts its modulatory effects on HCN channels has not yet been elucidated. In addition, there is currently no evidence for P2X7 overexpression in RP retina or RGCs. Therefore, it’s not clear what specific role Figure 1 serves in the manuscript.”

Response:

We agree that the mechanistic pathways illustrated in Fig. 1 remain incompletely defined. To address this concern, in the Introduction, the figure is reframed as a “conceptual model” and the legend has been revised to clarify explicitly that Fig. 1 represents a hypothesis-generating conceptual schematic informed by prior preclinical work on azobenzene photoswitches. The legend now states that the precise molecular gating mechanisms, the role of specific ion channels and the relevance of these pathways in human retinitis pigmentosa remain incompletely defined and are not implied as established. The purpose of Fig. 1 is therefore to provide conceptual context for potential mechanisms rather than to assert a definitive pathway.

Change made in manuscript:

The Fig. 1 legend has been revised to emphasise its conceptual and hypothesis-generating nature and to clarify that the mechanistic pathways shown are not established in human disease.

Comment 2

“In response to Q2, the authors emphasized that the current study (focusing on safety) was not designed to establish a retinal irradiance threshold for functional responses in humans... Ideally, clinical reports using FST is expected...”

Response:

We appreciate this comment and agree that full-field stimulus testing (FST) is an important method for quantifying global retinal light sensitivity. To clarify the rationale for its absence, we have added text to the Methods noting that FST was not included as a prespecified outcome because this Phase 1 study was not designed to establish irradiance thresholds or efficacy endpoints. The study was designed primarily to evaluate safety and feasibility, and functional measures were exploratory.

Change made in manuscript:

Added to Discussion:

“Full-field stimulus testing, while valuable for quantifying global retinal light sensitivity, was not included as a prespecified outcome because this study was not designed to establish irradiance thresholds or efficacy endpoints.”

Reviewer #3

No comments were provided. We thank the reviewer for their assessment.